# Gait Biomechanical Parameters Related to Falls in the Elderly: A Systematic Review

Jullyanne Silva [1], Tiago Atalaia [2], João Abrantes [3] and Pedro Aleixo [1,*]

1   Centro de Investigação em Desporto, Educação Física, Exercício e Saúde (CIDEFES), Universidade Lusófona, 1749-024 Lisbon, Portugal; jullyanne_silva@hotmail.com
2   Centro de Investigação em Ciências e Tecnologias da Saúde (CrossI&D), Escola Superior de Saúde da Cruz Vermelha Portuguesa-Lisboa, Physiotherapy, 1300-125 Lisbon, Portugal; tatalaia@esscvp.eu
3   Centre for Research in Applied Communication, Culture, and New Technologies (CICANT), Universidade Lusófona, 1749-024 Lisbon, Portugal; joao.mcs.abrantes@ulusofona.pt
*   Correspondence: pedro.aleixo@ulusofona.pt

**Abstract:** According to the World Health Organization, one-third of elderly people aged 65 or over fall annually, and this number increases after 70. Several gait biomechanical parameters were associated with a history of falls. This study aimed to conduct a systematic review to identify and describe the gait biomechanical parameters related to falls in the elderly. MEDLINE Complete, Cochrane, Web of Science, and CINAHL Complete were searched for articles on 22 November 2023, using the following search sentence: (gait) AND (fall*) AND ((elder*) OR (old*) OR (senior*)) AND ((kinematic*) OR (kinetic*) OR (biomechanic*) OR (electromyogram*) OR (emg) OR (motion analysis*) OR (plantar pressure)). This search identified 13,988 studies. From these, 96 were selected. Gait speed, stride/step length, and double support phase are gait biomechanical parameters that differentiate fallers from non-fallers. Fallers also tended to exhibit higher variability in gait biomechanical parameters, namely the minimum foot/toe clearance variability. Although the studies were scarce, differences between fallers and non-fallers were found regarding lower limb muscular activity and joint biomechanics. Due to the scarce literature and contradictory results among studies, it is complex to draw clear conclusions for parameters related to postural stability. Minimum foot/toe clearance, step width, and knee kinematics did not differentiate fallers from non-fallers.

**Keywords:** gait; falls; spatiotemporal parameters; kinematics; kinetics; systematic review

## 1. Introduction

According to the World Health Organization [1], the occurrence of falls in the elderly population is an important public health problem, representing the second leading cause of unintentional injury deaths worldwide. Several individual consequences of falls are described in the literature: reduced quality of life [2], psychological effects, such as increased fear of falling and loss of confidence [3], and fractures [4]. The elderly are the group with the highest incidence and worst consequences of falls, which increase with aging [5]. On the other hand, it is also important to highlight the economic burden of falls for families, communities, and society, e.g., falls among the elderly in the United States of America resulted in substantial medical costs [6].

Falls in the elderly are dependent on complex multifactorial risks such as (1) intrinsic risks, e.g., muscle weakness, stability disorders, functional and cognitive impairment, and visual deficits; (2) extrinsic risks, e.g., prescription of four or more medications; and (3) environmental risks, e.g., poor lighting or rugs that slide [7]. Focusing our attention on intrinsic factors, it is important to highlight that the subjects' functional capacity and motor control play an important role in falls. This is particularly important during walking, one of the activities of daily life in which falls are most prevalent [8]. Moreover, the increment of knowledge about gait biomechanics related to falls (i.e., objective data of functional

capacity and motor control) may help in the identification of subjects that present a high risk of falls and also in the development of interventions to decrease this risk [9]. In this context, tripping and slipping are the most frequent causes of falls during gait [10]. Regarding tripping, the minimum toe clearance (MTC) emerged as one gait biomechanical parameter related to this kind of fall. A previous systematic review [11] addressed this issue and found no differences between fallers and non-fallers regarding MTC, although the literature found was scarce. Furthermore, this systematic review also found that fallers had greater variability in MTC. Naturally, the biomechanical parameters related to slipping are different. In this way, the heel anteroposterior (AP) velocity at heel strike has been a biomechanical gait parameter related to slips [12]. Finally, the gait spatiotemporal parameters were also associated with a history of falls, namely gait speed and cadence [13,14]; stride length, double support phase duration, and variability in the stride length and swing time [14]; and variability in the step length and double-support phase [13].

Systematic reviews can synthesize the state of knowledge on a specific topic, allowing the identification of knowledge gaps that can constitute priorities for future research [15]. In our study, the synthesis of knowledge on the gait biomechanical parameters related to falls in the elderly may be useful for the stratification of severity (i.e., risk of fall) in future research and also helpful in developing more tailored and suitable assessments and interventions for this population [16]. Nonetheless, to the best of our knowledge, no systematic review has approached this issue. Therefore, the aim of this systematic review was to identify and describe the gait biomechanical parameters related to falls in elderly populations.

## 2. Materials and Methods

This systematic review was developed in accordance with the Preferred Items for Reporting for Systematic Reviews and Meta-Analysis (PRISMA) statement [17]. This review was registered in PROSPERO (ID–CRD42021271511).

### 2.1. Eligibility Criteria

The present review included studies according to the following inclusion criteria: (1) articles comparing elderly fallers and non-fallers on data from gait biomechanical analyses (the distinction between fallers and non-fallers can occur through retrospective studies, based on a previous history of falls, and prospective studies based on a follow-up of a fall's occurrence); (2) articles that during their methodological set-up induced falls and compared the elderly who fell with the elderly who recovered, regarding gait biomechanical data. In the scope of this study, gait biomechanical analyses also comprised analyses on uneven terrain. Furthermore, data from biomechanical gait analyses included spatiotemporal, kinematic, kinetic, electromyography (EMG), and plantar pressure data. The WHO definition of elderly was also considered, i.e., subjects aged 60 or over [18]. The following exclusion criteria were also defined: (1) articles assessing subjects under 60 years (or the mean age of the subjects less one standard deviation lower than 60 years); (2) articles assessing subjects that use any walking aid, with neurological or osteoarticular disease (e.g., stroke, Parkinson's disease, polyneuropathy, osteoarthritis, or rheumatoid arthritis), with dementia, who are amputees, or who are blind; (3) articles assessing subjects during dual-task, ascending or descending stairs, turning, or obstacle-crossing; and (4) case reports, reviews, or dissertations. No restrictions were imposed regarding language or publication date.

### 2.2. Search Strategy and Selection Process

Systematic review was conducted independently by two researchers (J.S. and P.A.), using the following protocol: (1) a comprehensive search of articles was made on MED-LINE Complete, Cochrane, Web of Science, and CINAHL Complete for articles published until 22 November 2023, using the following search sentence- (gait) AND (fall*) AND ((elder*) OR (old*) OR (senior*)) AND ((kinematic*) OR (kinetic*) OR (biomechanic*) OR

(electromyogram*) OR (emg) OR (motion analysis*) OR (plantar pressure)); (2) exclusion of duplicates—Mendeley was used to manage all references, removing duplicates; (3) selection of articles by title and abstract; (4) screening of articles by analyzing the full text; and (5) hand search for relevant articles. In the third and fourth points of the protocol, a third reviewer (T.A.) would be consulted if there were any disagreements between the two reviewers.

### 2.3. Data Extraction and Synthesis

Data were extracted by one reviewer (J.S.) using a predefined form: (1) authors and year of publication; (2) inclusion and exclusion criteria and definition of fall; (3) sample characteristics (sample size and sociodemographic data—age, gender, and type of population); (4) gait assessment; and (5) results (gait biomechanical parameters related or not related to falls). All information was cross-checked by a second reviewer (P.A.).

### 2.4. Risk of Bias Assessment

In this systematic review, the Quality Assessment Tool for Quantitative Studies, developed by the Effective Public Health Practice Project [19], was used to assess the methodological quality of studies. This assessment focused on 6 domains: (1) selection bias; (2) study design; (3) confounders; (4) blinding; (5) data collection method; and (6) withdrawals and dropouts. Each domain was evaluated with the following classification: "1" corresponds to a strong report in that domain; "2" corresponds to a reasonable report in that domain; and "3" corresponds to a weak report in that domain. The articles that met the inclusion criteria were assessed independently by two researchers (J.S. and P.A.). Any disagreements were resolved with a consensus discussion between them. If disagreements remained after discussion, a third reviewer was consulted (T.A.).

### 3. Results

A total of 13,988 records were found from the following databases: MEDLINE Complete and CINAHL Complete (7144), Cochrane (887), and Web of Science (5957). Additional articles were also found by manual searching (12). Removal of duplicates resulted in 9425 eligible articles. After this first selection, 8512 articles were excluded by determining that their titles and abstracts were not relevant or did not meet the inclusion criteria. In this way, the full text of 913 records was reviewed. The eligibility process resulted in the inclusion of 96 articles in this systematic review. A flow diagram summarizing this selection process is shown in Figure 1.

### 3.1. Characteristics of the Selected Studies

This review included 96 studies: 86 studies [14,20–104] comparing elderly fallers and non-fallers (Table 1) and 12 studies [103–114] that induced falls during their methodological set-up and compared the elderly who fell with the elderly who recovered (Table 2).

**Table 1.** Characteristics and data of the studies that compared fallers' and non-fallers' gait.

| Study | Inclusion and/or Exclusion Criteria Definition of Fall | Sample Characteristics | Gait Assessment | Gait Parameters Related to Falls (Fallers vs. Non-Fallers) | Gait Parameters Not Related to Falls (Fallers vs. Non-Fallers) |
|---|---|---|---|---|---|
| Heitmann et al., 1989 [20] | Exclusion criteria: Parkinson's disease; multiple sclerosis; or residual effects from a stroke. Inclusion criteria: able to walk 90 feet without an assistive device and to be independent in activities of daily living. Definition of fall: not reported. | Community-dwelling elderly women: 26 fallers (≥1 fall in past year; 75.1 ± 7.7 years). 84 non-fallers (73.1 ± 7.0 years). | Subjects walked on paper walkways making ink prints for step-width measurements. 3 trials were performed; the best one was used for analysis. | | Step width (cm): 7.44 vs. 6.54; step width variability (SD; cm): 3.60 vs. 3.35. |
| Gehlsen & Whaley, 1990 [21] | Exclusion criteria: uncontrolled hypertension; angina pectoris; recent myocardial infarction; or disabling injury to legs and back. Definition of fall: not reported. | Community-dwelling elderly: 25 fallers (≥1 fall in past 10 months; 7 males; 72.4 ± 4.7 years). 30 non-fallers (12 males; 71.3 ± 4.4 years). | Subjects walked on a treadmill at 4 km/h and 6 km/h. They were filmed by two cameras (64 Hz; sagittal and frontal planes). MTC was analyzed. | 6 km/h: heel width (cm): 7.39 vs. 6.41. | 4 km/h: heel width (cm): 7.77 vs. 7.19. 4 km/h and 6 km/h: stride length (m): 0.58 vs. 0.59; 0.73 vs. 0.72; MTC (cm): 1.15 vs. 0.70; 0.77 vs. 0.68; single support phase (s): 0.49 vs. 0.50; 0.45 vs. 0.44; double support phase (s): 0.14 vs. 0.15; 0.11 vs. 0.11; swing phase (s): 0.35 vs. 0.35; 0.34 vs. 0.34; cadence (stride/s): 2.05 vs. 2.01; 2.26 vs. 2.21; hip displacement (°): 65–104 vs. 64–105; knee displacement (°): 9–62 vs. 8–63; 7–63 vs. 8–62; ankle displacement (°): 98–120 vs. 97–120; 103–125 vs. 99–122. |
| Feltner et al., 1994 [22] | Exclusion and inclusion criteria: not reported. Definition of fall: event that results in a subject coming to rest inadvertently on the ground. | Community-dwelling elderly women: 6 fallers (≥1 fall in past year; 71.7 ± 2.6 years). 11 non-fallers (74.4 ± 1.7 years). | Subjects walked at their preferred gait speed across an 8.2 m walkway filmed by two cameras (60 Hz; sagittal and frontal planes). At least 3 trials were collected, and the trial with a complete stride in the side view camera footage was used for analysis. AP and ML velocities of CoM were calculated. | | Stride length (m): 1.12 vs. 1.16; right and left step length (m): 0.57 vs. 0.60, 0.54 vs. 0.56; step width (m): 0.22 vs. 0.22; stride time (s): 1.05 vs. 1.02; right and left step time (s): 0.59 vs. 0.53, 0.46 vs. 0.50; single support (%): 69.8 vs. 68.5; swing phase (%): 31.1 vs. 32.9; CoM AP velocity (m/s): 1.08 vs. 1.14; CoM ML velocity (m/s): −0.19 vs. −0.20; minimum and maximum hip position (°): −6 vs. −10; 26 vs. 26; minimum and maximum knee position (°): 175 vs. 175; 118 vs. 116; minimum and maximum ankle position (°): 12 vs. 8; −12 vs. −14; width of the base of support (m): 0.14 vs. 0.15. |

**Table 1.** *Cont.*

| Study | Inclusion and/or Exclusion Criteria Definition of Fall | Sample Characteristics | Gait Assessment | Gait Parameters Related to Falls (Fallers vs. Non-Fallers) | Gait Parameters Not Related to Falls (Fallers vs. Non-Fallers) |
|---|---|---|---|---|---|
| Wolfson et al., 1995 [23] | Exclusion criteria: terminal illness; severe dementia; non-ambulatory status; required use of a walker; amputations; severe arthritis; major impairment due to neurologic disease; or episodes of loss of consciousness. Definition of fall: not reported. | Community-dwelling elderly: 18 fallers ($\geq$2 falls in past year; mean age 82.2 years). 15 non-fallers (mean age 84.6 years). | Not reported. | Gait speed (m/s): 0.37 vs. 0.64; stride length (m): 0.53 vs. 0.82. | |
| Maki, 1997 [24] | Inclusion criteria: able to walk 10 m with or without a walking aid and understand verbal instructions. Definition of fall: event that results in a subject coming to rest inadvertently on the ground. | Community-dwelling elderly: 43 fallers ($\geq$1 fall in 1-year follow-up; 8 males; 82.8 $\pm$ 6.2 years). 32 non-fallers (6 males; 81.0 $\pm$ 6.7 years). | Subjects walked with their own footwear at their preferred gait speed across an 8 m walkway. Four trials were filmed but only the last two were included in the analysis. | Stride length variability (SD): higher values in fallers; double support phase variability (SD): higher values in fallers; gait speed variability (SD): higher values in fallers. | Stride length; stride time; double support phase; gait speed; stride width variability (SD); stride time variability (SD). |
| Lee & Kerrigan, 1999 [25] | Exclusion criteria: musculoskeletal, neurological, or cardiac diseases. Definition of fall: event that results in a subject coming to rest inadvertently on the ground or other lower level. | Community-dwelling elderly: 15 fallers ($\geq$2 falls in past 6 months; 7 males; 77.0 $\pm$ 9.0 years). 15 non-fallers (8 males; 75.0 $\pm$ 5.0 years). | Subjects walked barefoot or with their shoes at a preferred gait speed across a 30-foot walkway. Kinematic data (using a 4-camera optoelectronic motion analysis system at 100 Hz) and ground reaction forces (using 2 force plates) were collected during 3 trials. | Gait speed (m/s): 0.41 vs. 0.82; cadence (steps/s): 86 vs. 111; step length (m): 0.24 vs. 0.40; hip flexion moment (Nm/kg): 0.96 vs. 0.44; hip adduction moment (Nm/kg): 1.49 vs. 0.69; knee varus moment (Nm/kg): 0.86 vs. 0.33; knee extension moment (Nm/kg): 0.40 vs. 0.21; ankle dorsiflexion moment (Nm/kg): 1.59 vs. 0.91; ankle plantarflexion moment (Nm/kg): 0.075 vs. 0.139; ankle eversion moment (Nm/kg): 0.43 vs. 0.13; knee power absorption (W/kg): 0.98 vs. 1.66; ankle power absorption (W/kg): 0.76 vs. 0.41. | Hip extension moment (Nm/kg): 0.67 vs. 0.74; knee flexion moment (Nm/kg): 0.59 vs. 0.46; ankle inversion moment (Nm/kg): 014 vs. 0.07; hip power generation (W/kg): 1.23 vs. 1.18; knee power generation (W/kg): 0.54 vs. 0.55; ankle power generation (W/kg): 2.03 vs. 2.04; hip power absorption (W/kg): 0.44 vs. 0.61. |
| Nelson et al., 1999 [26] | Inclusion criteria: independent subjects. Definition of fall: not reported. | Community-dwelling elderly: 11 fallers (1 male; 79.4 $\pm$ 8.7 years). 13 non-fallers (4 males; 80.1 $\pm$ 6.0 years). | Subjects walked on an electronic walkway at their preferred gait speed and completed 4 trials. | Gait speed (m/s): 0.82 vs. 1.25; left and right step times (s): 0.61 vs. 0.53, 0.60 vs. 0.52; left and right heel to heel base of support (cm): 12.5 vs. 9.7, 12.4 vs. 9.6; left and right double support phase (%): 35.0 vs. 26.0, 34.0 vs. 26.0. | |

**Table 1.** *Cont.*

| Study | Inclusion and/or Exclusion Criteria Definition of Fall | Sample Characteristics | Gait Assessment | Gait Parameters Related to Falls (Fallers vs. Non-Fallers) | Gait Parameters Not Related to Falls (Fallers vs. Non-Fallers) |
|---|---|---|---|---|---|
| Wall et al., 2000 [27] | Exclusion and inclusion criteria: not reported. Definition of fall: not reported. | Community-dwelling elderly: 10 fallers (≥2 falls in past 2 years; 75.8 ± 9.3 years). 10 non-fallers (72.7 ± 4.0 years). | Gait was assessed during an expanded Timed Up and Go test. A 10 m walkway was used. A stopwatch recorded the intervals between each phase. | Front walk: gait speed (m/s): 0.81 vs. 1.23. Return walk: gait speed (m/s): 0.78 vs. 1.23. | |
| Hausdorff et al., 2001 [28] | Exclusion criteria: unable to follow simple instructions; nursing home residents; or life expectancy of less than 1 year. Definition of fall: not reported. | Community-dwelling elderly (16 males; 80.3 ± 5.9 years): 20 fallers (≥1 fall in 1-year follow-up). 32 non-fallers. | Subjects walked at their preferred gait speed for up to 6 min, wearing force-sensitive insoles that measured the gait rhythm on a stride-to-stride basis. | Stride time variability (SD; s): 0.11 vs. 0.05; swing time variability (SD; s): 0.04 vs. 0.03. | Gait speed (m/s): 0.71 vs. 0.91 (statistical tendency for difference, $p = 0.078$). |
| Kerrigan et al., 2001 [30] Kerrigan et al., 2000 [29] | Exclusion criteria: acute medical illness; diagnosis or symptoms of unstable angina or congestive heart failure; pulmonary disease diagnosis or symptoms; neurologic disorders that impair mobility; major orthopedic diagnosis in the lower back, pelvis, or lower extremities; or active joint or musculoskeletal pain. Additional exclusion criteria for fallers: falls secondary to syncope, acute illness, or other specific causes including metabolic disorders; medication side effects, true vertigo; or neurologic or lower extremity orthopedic diagnoses. Definition of fall: event that results in a subject coming to rest inadvertently on the ground or other lower level. | Community-dwelling elderly: 16 fallers (≥2 falls in last 6 months; 8 males; 77.0 ± 7.8 years). 23 non-fallers (10 males; 73.2 ± 5.6 years). | Subjects walked barefoot at their preferred and fast gait speed across a 10 m walkway. Kinematic data were collected during 3 trials using an optoelectronic motion analysis system at 100 Hz and ground reaction forces using 2 force plates. | Preferred gait speed: gait speed (m/s): 0.89 vs. 1.21; stride length (m): 0.98 vs. 1.22; hip flexion moment (Nm/kg): 0.53 vs. 0.38; hip extension moment (Nm/kg): 0.22 vs. 0.54; hip power absorption (W/kg): 0.13 vs. 0.40; hip power generation during pre-swing (W/kg): 0.43 vs. 0.92; hip adduction moment (Nm/kg): 0.47 vs. 0.57; knee flexion moment during mid-stance (Nm/kg): 0.15 vs. 0.27; knee flexion moment during pre-swing (Nm/kg): 0.07 vs. 0.24; knee power absorption during pre-swing (W/kg): 0.31 vs. 1.29; ankle power generation during pre-swing (W/kg): 1.10 vs. 1.74; hip extension (°): 11 vs. 14. Fast gait speed: gait speed (m/s): 1.34 vs. 1.57; stride length (m): 1.17 vs. 1.34; hip extension (°): 12 vs. 14. | Preferred speed: cadence (steps/min): 107 vs. 120; hip flexion moment during swing (Nm/kg): 0.08 vs. 0.11; hip power generation during loading response (W/kg): 0.50 vs. 0.50; hip abduction moment (Nm/kg): 0.08 vs. 0.07; hip internal rotation moment (Nm/kg): 0.14 vs. 0.14; hip external rotation moment (Nm/kg): 0.09 vs. 0.12; knee extension moment during terminal stance (Nm/kg): 0.16 vs. 0.16; knee power absorption during loading response (W/kg): 0.14 vs. 0.27; knee power generation during mid-stance (W/kg): 0.25 vs. 0.35; knee varus moment (Nm/kg): 0.25 vs. 0.27; knee valgus moment (Nm/kg): 0.02 vs. 0.02; knee internal rotation moment (Nm/kg): 0.14 vs. 0.13; knee external rotation moment (Nm/kg): 0.10 vs. 0.11; ankle plantarflexion moment (Nm/kg): 0.06 vs. 0.09; ankle dorsiflexion moment (Nm/kg): 0.73 vs. 0.75; ankle power absorption (W/kg): 0.43 vs. 0.44; ankle inversion moment (Nm/kg): 0.02 vs. 0.05; ankle eversion moment (Nm/kg): 0.18 vs. 0.11; ankle internal rotation |

**Table 1.** *Cont.*

| Study | Inclusion and/or Exclusion Criteria Definition of Fall | Sample Characteristics | Gait Assessment | Gait Parameters Related to Falls (Fallers vs. Non-Fallers) | Gait Parameters Not Related to Falls (Fallers vs. Non-Fallers) |
|---|---|---|---|---|---|
| | | | | | moment (Nm/kg): 0.17 vs. 0.16; ankle external rotation moment (Nm/kg): 0.09 vs. 0.09; hip flexion (°): 21 vs. 26; knee flexion during stance (°): 11 vs. 17; knee extension during stance (°): 2 vs. 2; knee flexion during swing (°): 52 vs. 58; knee extension during swing (°): 2 vs. 3; ankle plantarflexion during initial stance (°): 8 vs. 8; ankle dorsiflexion during mid-stance (°): 8 vs. 9; ankle plantarflexion (°): 14 vs. 15; ankle dorsiflexion during swing (°): 2 vs. 2; peak anterior pelvic tilt (°): 3 vs. 3. Fast gait speed: cadence (steps/min): 138 vs. 140; hip flexion (°): 25 vs. 30; knee flexion during stance (°): 16 vs. 21; knee extension during stance (°): 3 vs. 2; knee flexion during swing (°): 55 vs. 61; knee extension during swing (°): 3 vs. 6; ankle plantarflexion initial during stance (°): 8 vs. 7; ankle dorsiflexion during mid-stance (°): 6 vs. 7; ankle plantarflexion (°): 14 vs. 16; ankle dorsiflexion during swing (°): 1 vs. 2; peak anterior pelvic tilt (°): 4 vs. 4. |
| Kemoun et al., 2002 [31] | Exclusion and inclusion criteria: not reported. Definition of fall: unexpected event when a subject falls to the ground from the same or an upper level (including falls on stairs and onto a piece of furniture). | Community-dwelling elderly (66.7 ± 4.8 years): 16 fallers (≥1 fall in 1-year follow-up; 12 males). 38 non-fallers (26 males). | Subjects walked barefoot at their preferred gait speed across a 10 m walkway. Kinematic data were collected during 5 trials using a 5-camera optoelectronic motion analysis system at 50 Hz and ground reaction force using two integrated force platforms at 250 Hz. | Gait speed (m/s): 0.96 vs. 1.29; double support phase (%): 27.8 vs. 23.2; ankle moment peak (Nm/kg): 25 vs. 23; ankle plantarflexion during second double support (°): 19 vs. 23; ankle dorsiflexion at beginning of swing (°): 7 vs. 13; hip power variation (W/kg): 1.02 vs. 2.04; hip moment peak (Nm/kg): −0.54 vs. −0.97; hip moment variation (Nm/kg): 0.88 vs. 1.60; hip displacement (°): 40 vs. 47. | Cadence (step/min): 99 vs. 108 (statistical tendency for difference, $p = 0.059$); stride length (m): 1.12 vs. 1.31; step length (m): 0.57 vs. 0.65; stride time (s): 1.20 vs. 1.11 (statistical tendency for difference, $p = 0.058$); single support phase (%): 37 vs. 38.2; step time (%): 49.3 vs. 50.0; single support start (%): 13.5 vs. 13.6; double support start (%): 50.7 vs. 50.0; swing start (%): 64.6 vs. 62.1 (statistical tendency for difference, $p = 0.077$); ankle power peak (W/kg): 2.5 vs. 3.1; ankle moment peak (Nm/kg): 1.58 vs. 1.54; knee power peak (W/kg): −0.81 vs. −1.35; knee power variation (W/kg): 0.91 vs. 1.42; knee moment peak (Nm/kg): −0.17 vs. −0.04; |

**Table 1.** *Cont.*

| Study | Inclusion and/or Exclusion Criteria Definition of Fall | Sample Characteristics | Gait Assessment | Gait Parameters Related to Falls (Fallers vs. Non-Fallers) | Gait Parameters Not Related to Falls (Fallers vs. Non-Fallers) |
|---|---|---|---|---|---|
| | | | | | knee moment variation (Nm/kg): 0.74 vs. 0.74; knee position peak (°): 62 vs. 63; knee displacement (°): 50 vs. 52; hip power peak (W/kg): −0.93 vs. −1.23; hip position peak (°): 57 vs. 61. |
| Auvinet et al., 2003 [32] | Inclusion criteria (fallers): recently hospitalized due to falls; living at home; and no pelvic or leg length asymmetries. Inclusion criteria (non-fallers): no history of musculoskeletal, neurological, or gait disorder; living at home; and no marked pelvic asymmetry or leg length differences. Definition of fall: not reported. | Community-dwelling elderly: 20 fallers (≥1 fall in past year; 2 males; 80.7 ± 5.2 years). 33 non-fallers (18 males; 77.2 ± 6.5 years). | Subjects walked at their preferred gait speed across a 10 m walkway using their own shoes. Gait parameters were collected using an accelerometer sensor system (50 Hz). | Gait speed (m/s): 0.73 vs. 1.24; stride length (m): 0.86 vs. 1.28; stride frequency-cadence (Hz): 0.86 vs. 0.97; stride symmetry: 173 vs. 211. | |
| Mbourou et al., 2003 [33] | Exclusion criteria: Parkinson's disease or Alzheimer's disease. Inclusion criteria (fallers): living in a nursing home. Definition of fall: not reported. | Elderly: 9 fallers (≥1 fall in past year; mean age 80.0 years, range 74.0–91.0). 8 non-fallers (mean age 73.0 years, range 66.0–82.0). | Subjects were asked to initiate gait and walk at least 3 strides. The length of the first step and subsequent strides were collected using transducers. Gait parameters were derived from the displacement signal obtained from each foot. More than 10 trials were collected. | First step length (m): 0.30 vs. 0.53; first step length variability (SD; m): 0.13 vs. 0.06; first double support phase (%): 32 vs. 22; second stride length (m): 0.68 vs. 0.92; second stride length variability (SD; m): 0.10 vs. 0.05; double support phase for subsequent strides (%): 37 vs. 32. | |
| Pijnappels et al., 2005 [103] | Exclusion and inclusion criteria: not reported. Definition of fall: when the vertical force in the ropes exceeded 200 N during trials when one obstacle appeared from the ground unexpectedly to catch the subject's swing limb. | Community-dwelling elderly: 7 fallers (when the vertical force in the ropes exceeded 200 N during trip trials; 1 male; 67.9 ± 2.6 years). 4 non-fallers (3 males; 66.5 ± 3.3 years). | Subjects walked at preferred gait speed over a platform. Kinematic and ground reaction force data were collected using a 4-camera optoelectronic motion analysis system and a force plate (100 Hz). | | Stance time; double support time. |
| Chiba et al., 2005 [34] | Exclusion criteria: Mini Mental Status Examination score < 24; arthritis in lower limbs; back, knee, or hip chronic pain; Parkinson's disease; Ménière's syndrome; cerebellar signs; or peripheral neuropathy under standard neuropsychological assessment. Inclusion criteria: being medically stable; | Community-dwelling elderly: 25 fallers (≥2 falls in past year; 11 males; 76.0 ± 6.6 years). 31 non-fallers (11 males; 74.9 ± 7.2 years). | Subjects walked barefoot or with their own shoes on a 6 m walkway. During 2 continuous trials, kinematic data were collected using a 5-camera optoelectronic motion analysis system (60 Hz). MTC was analyzed. | Gait speed (m/s): 0.66 vs. 0.99; stride length (m): 0.77 vs. 1.06; stride time (s): 1.20 vs. 1.08; MTC (mm): 12.0 vs. 15.2; MTC variability (CV; %): 29 vs. 25; maximum foot angle with ground (°): 7.4 vs. 14.3; variability in the maximum foot angle with ground (CV; %): | |

**Table 1.** *Cont.*

| Study | Inclusion and/or Exclusion Criteria Definition of Fall | Sample Characteristics | Gait Assessment | Gait Parameters Related to Falls (Fallers vs. Non-Fallers) | Gait Parameters Not Related to Falls (Fallers vs. Non-Fallers) |
|---|---|---|---|---|---|
| | comprehending the nature of the study and our instructions; and being able to stand up and walk independently without an assistance device. Definition of fall: a sudden unintentional change in position causing a subject to land at a lower level or ground. | | | 34 vs. 19; maximal ML displacement of trunk center (mm): 0.23 vs. 0.18; variability in the maximal ML displacement of trunk center (CV; %): 6 vs. 1. | |
| Barak et al., 2006 [35] | Exclusion criteria: cardiopulmonary, musculoskeletal, somatosensory, or neurological disorders or severe visual and vestibular loss. Definition of fall: not reported. | Community-dwelling elderly: 21 fallers ($\geq$1 fall in last 6 months; 8 males; 73.8 $\pm$ 6.4 years). 27 non-fallers (14 males; 72.1 $\pm$ 4.9 years). | Subjects walked at their preferred gait speed; treadmill speed was gradually increased from 0.18 m/s to 1.52 m/s in steps of 0.225 m/s and then decreased. During 1 min in each step, kinematic data were collected using an optoelectronic motion analysis system (100 Hz). CoM was calculated. | Effects of gait speed: cadence: in all gait speeds; stride length: only in 1.3 m/s gait speed; CoM lateral sway: from 1.07 m/s gait speed; ankle plantarflexion: from 1.07 m/s gait speed; hip extension: from 0.85 m/s gait speed; hip flexion: from 0.85 m/s gait speed; hip flexion variability: only in 1.52 m/s gait speed. Effects of stride frequency: stride length: in 1.1 and 1.2 stride frequencies; lateral body sway: in 0.6–1.1 stride frequencies; hip extension: in 0.5–1.0 stride frequencies; hip extension variability: in 0.9–1.2 stride frequencies; hip flexion: in 1.0 stride frequency; hip flexion variability: in 0.9–1.2 stride frequency. | Effects of stride frequency: ankle plantarflexion. |
| Toulotte et al., 2006 [36] | Exclusion criteria: lower limb fracture or surgery; use of walking aid or foot orthosis; cognitive disorders; auditory, ocular, or vestibular problems; head trauma with/without loss of consciousness; stroke; carpal tunnel syndrome; or sores on lower limbs or corns. Inclusion criteria: $\geq$60 years and stable medical treatment (for at least 3 months). Definition of fall: any event that led to an unplanned contact with a supporting surface. | Community-dwelling elderly women: 21 fallers ($\geq$1 fall in past 2 years; 70.4 $\pm$ 6.4 years). 19 non-fallers (67.0 $\pm$ 4.8 years). | Subjects walked at preferred gait speed across a 10 m walkway. During 10 trials, kinematic data were collected using an optoelectronic motion analysis system (50 Hz) and three force plates (250 Hz). | | Cadence (steps/min): 116 vs. 119; gait speed (m/s): 1.08 vs. 1.12; stride time (s): 1.04 vs. 1.02; step time (s): 0.53 vs. 0.51; single support time (s): 0.48 vs. 0.48; stride length (m): 1.13 vs. 1.09; step length (m): 0.56 vs. 0.60. |

**Table 1.** *Cont.*

| Study | Inclusion and/or Exclusion Criteria Definition of Fall | Sample Characteristics | Gait Assessment | Gait Parameters Related to Falls (Fallers vs. Non-Fallers) | Gait Parameters Not Related to Falls (Fallers vs. Non-Fallers) |
|---|---|---|---|---|---|
| Karmakar et al., 2012 [38] Karmakar et al., 2007 [37] | Exclusion and inclusion criteria: not reported. Definition of fall: not reported. | Community-dwelling elderly women: 10 fallers ($\geq$1 fall in past year; 72.2 $\pm$ 3.1 years). 27 non-fallers (69.1 $\pm$ 5.1 years). | Subjects walked on a treadmill at preferred gait speed. During the first 500 continuous gait cycles, MTC data were collected using a 2D motion analysis system and analyzed by an ApEn and a SampEn. ApEn was calculated with m = 3 and r from 0 to 90% of the calculated SD. SampEn was calculated with m varying from 2 to 4 and r from 0 to 90% of the calculated SD. | Gait speed (m/s): 0.91 vs. 1.29; MTC (cm): 2.02 vs. 1.25; MTC variability (SD; cm): 0.47 vs. 0.32. For r < 0.26 * SD, the mean MTC ApEn of fallers was higher than non-fallers. For r $\geq$ 0.26 * SD, the mean MTC ApEn of fallers was smaller than non-fallers. MTC SampEn values of fallers were lower compared to non-fallers for all m and r. | |
| Newstead et al., 2007 [39] | Exclusion criteria: neurological or orthopedic conditions. Inclusion criteria: $\geq$60 years; be able to walk 1 mile nonstop; and free of neurological or orthopedic impairments. Definition of fall: not reported. | Community-dwelling elderly: 18 fallers ($\geq$1 fall in past year; 3 males; 78.1 $\pm$ 7.2 years). 30 non-fallers (6 males; 75.8 $\pm$ 5.1 years). | Subjects walked 5–7 trials at three different gait speeds (slow, preferred, and fast) across a 10 m walkway using laced walking shoes. Spatiotemporal data were collected using a 6-camera optoelectronic motion analysis system (60 Hz) and four force plates (250 Hz). | Preferred gait speed: gait speed (lower fallers); cadence (lower fallers); step length (lower fallers); stride length (lower fallers); single support time (lower fallers); double support time (higher fallers). Fast gait speed: stride length (lower fallers); cadence (lower fallers); gait speed (lower fallers). | Slow gait speed: gait speed; cadence; step length; stride length; single support time; double support time. Fast gait speed: step length; single support time; double support time. |
| Barrett et al., 2008 [40] | Exclusion criteria: limited pulmonary and cardiac function; use of pacemakers; or cognitive impairment. Inclusion criteria: independent ambulation for at least 6 m. Definition of fall: subject who had fallen from vertical to horizontal. | Community-dwelling elderly: 9 fallers ($\geq$2 falls in past year; 5 males; 76.0 $\pm$ 5.0 years). 10 non-fallers (5 males 69.0 $\pm$ 5.0 years). | Subjects walked at preferred gait speed over a 6 m walkway. Gait events were detected using footswitches embedded in the left shoe. Three walks were recorded for analysis. | Stance time (higher fallers); stride time (higher fallers); stance phase (higher fallers); stance time variability (SD; higher fallers); stride time variability (SD; higher fallers). | Swing time; swing time variability (SD). |
| Khandoker et al., 2008 [41] | Exclusion and inclusion criteria: not reported. Definition of fall: not reported. | Community-dwelling elderly: 10 fallers ($\geq$1 fall; 72.2 $\pm$ 3.1 years). 14 non-fallers (71.0 $\pm$ 2.1 years). | Subjects walked 10–20 min on the treadmill. MTC data were collected using a 2D motion analysis system (50 Hz) and analyzed by ApEn and Poincaré Plot Indexes. ApEn was calculated with m = 3 and r from 0 to 90%. SampEn was calculated with m varying from 2 to 3 and r from 15%. | MTC ApEn values: 0.18 vs. 0.13; MFC variability (SD; cm): 0.48 vs. 0.35; Poincaré width: 0.72 vs. 0.51; Poincaré length: 1.15 vs. 0.89. | MFC (cm): 2.01 vs. 1.65; Poincaré width/Poincaré length: 0.64 vs. 0.64. |

**Table 1.** *Cont.*

| Study | Inclusion and/or Exclusion Criteria Definition of Fall | Sample Characteristics | Gait Assessment | Gait Parameters Related to Falls (Fallers vs. Non-Fallers) | Gait Parameters Not Related to Falls (Fallers vs. Non-Fallers) |
|---|---|---|---|---|---|
| Khandoker et al., 2008 [42] | Exclusion and inclusion criteria: not reported. Definition of fall: not reported. | Community-dwelling elderly women: 10 fallers ($\geq$1 fall in past year; 72.2 $\pm$ 3.1 years). 27 non-fallers (69.1 $\pm$ 5.1 years). | Subjects walked 10 min on a treadmill at preferred gait speed. MTC data were collected using a 2D motion analysis system (50 Hz). The following variability indices were quantified: Poincaré plot indices (SD1, SD2, SD1/SD2); wavelet-based multiscale exponent; and detrended fluctuation analysis exponent to investigate the presence of long-range correlations in MTC time series. | MTC (cm): 2.02 vs. 1.25. Wavelet-based multiscale exponent, SD1/SD2, and SD2 of critical MTC parameters were found to be potential markers to be able to reliably identify fallers from non-fallers. | |
| Lockhart & Liu, 2008 [43] | Exclusion and inclusion criteria: not reported. Definition of fall: not reported. | Community-dwelling elderly: 4 fallers ($\geq$1 fall in past 6 months; 70.1 $\pm$ 3.0 years). 4 non-fallers (71.3 $\pm$ 6.5 years). | Subjects walked for 1 min on a treadmill at their preferred gait speed. One dual-axial accelerometer was placed on the right anterior superior iliac spine (125 Hz). Maximum Lyapunov exponent was used to analyze these data. Two infrared-reflective markers were placed bilaterally on the heels for kinematic capture with a 6-camera optoelectronic motion analysis system (120 Hz). | Maximum Lyapunov exponent: 2.39 vs. 1.99; step length (m): 0.33 vs. 0.60; gait speed (m/s): 0.57 vs. 1.16. | Heel contact velocity (m/s): 0.32 vs. 0.43; step duration (s): 1.19 vs. 1.04. |
| Verghese et al., 2009 [14] | Exclusion criteria: severe audiovisual loss; bed-bound due to illness; or institutionalization. Definition of fall: unintentionally coming down on the floor or to a lower level, not due to a major intrinsic or extrinsic event. | Community-dwelling elderly (227 males; mean age 80.6 years): 226 fallers (115 fell once and 111 had recurrent falls; mean age 81.1 years). 371 non-fallers (mean age 80.1 years). | Subjects walked at preferred gait speed during 2 trials on a computerized walkway with embedded pressure sensors using comfortable footwear. Generalized estimating equations with a binomial distribution to model the probability of fall. | Slower gait speed (risk ratio per 10 cm/s decrease 1.069, 95% confidence interval: 1.001–1.142) is associated with a higher risk of falls. Predicted fall risk: swing phase (RR 1.406, 95% confidence interval 1.027–1.926); double-support phase (RR 1.165, 95% confidence interval: 1.026–1.321); swing time variability–CV (RR 1.007, 95% confidence interval: 1.004–1.010); stride length variability–CV (RR 1.076, 95% confidence interval: 1.030–1.111). | |
| Greany & Di Fabio, 2010 [44] | Inclusion criteria: $\geq$70 years; living at home; can walk at least 30 feet without stopping; Mini Mental Status Examination score > 23; corrected visual acuity of at least 20/70; and peripheral visual field of 30°. Definition of fall: unintentionally coming to rest on the ground. | Community-dwelling elderly: 12 fallers ($\geq$1 fall in past year; 3 males, 86.0 $\pm$ 4.8 years). 21 non-fallers (7 males, 81.0 $\pm$ 5.0 years). | Subjects walked quickly but safely along a walkway of four irregularly spaced stepping targets. During 6 trials, a video-based motion analysis system was used for collecting kinematic data. | | Maximum foot AP velocity (m/s): 1.91 vs. 2.05; maximum foot vertical velocity (m/s): 0.54 vs. 0.53; average foot AP velocity (m/s): 0.36 vs. 0.36; step time (s): 1.64 vs. 1.51; step length (m): 0.76 vs. 0.83; swing time (s): 0.67 vs. 0.68; double support time (s): 0.45 vs. 0.35. |

**Table 1.** *Cont.*

| Study | Inclusion and/or Exclusion Criteria Definition of Fall | Sample Characteristics | Gait Assessment | Gait Parameters Related to Falls (Fallers vs. Non-Fallers) | Gait Parameters Not Related to Falls (Fallers vs. Non-Fallers) |
|---|---|---|---|---|---|
| Greene et al., 2010 [45] | Inclusion criteria: ≥60 years; able to walk independently with or without help; and able to provide informed consent. Definition of fall: unexpected loss of stability resulting in coming to rest on the floor or an object below the knee level. | Community-dwelling elderly: 207 fallers (≥1 fall in past 5 years; 44 males; 74.0 ± 7.3 years). 142 non-fallers (59 males; 71.1 ± 6.9 years). | Gait was assessed during the Timed Up and Go test through two wearable tri-axial accelerometer sensors placed on each shank. | Cadence (steps/min): 99 vs. 108; double support (s): 0.4 vs. 0.5; step time (s): 0.7 vs. 0.6; minimum shank ML angular velocity (lower fallers); mean shank ML angular velocity (lower fallers); maximum shank ML angular velocity (lower fallers); minimum shank AP angular velocity (lower fallers); mean shank AP angular velocity (lower fallers); maximum shank AP angular velocity (lower fallers). | Stance time (s): 0.8 vs. 0.8; single support time (s): 0.8 vs. 0.8; stride time (s): 1.2 vs. 1.2; swing time (s): 0.5 vs. 0.5; single support time variability (CV; %): 22.9 vs. 21.1; double support variability (CV; %): 80.7 vs. 82.6; swing time variability (CV; s): 28.1 vs. 31.0; stride time variability (CV; s): 24.0 vs. 23.4; step time variability (CV; s): 42.0 vs. 40.3; stance time variability (CV; s): 43.3 vs. 45.0. |
| Mickle et al., 2010 [46] | Inclusion criteria: ≥60 years; living independently in the community; passed the Short Portable Mental Status Questionnaire; able to ambulate for at least 10 m with or without aid; no neurological diseases; and own transport to a testing venue in the community. Definition of fall: unintentionally coming to rest on the ground or other lower level, not as a result of a major intrinsic event (e.g., stroke). | Community-dwelling elderly: 107 fallers (≥1 fall in 1-year follow-up; 49 males; 71.6 years). 196 non-fallers (105 males; 71.2 years). | Five trials were recorded with a two-step gait initiation protocol at a preferred walking speed. A pressure platform was used. | Peak pressure (KPa): 776 vs. 699; pressure–time integral (KPa): 349 vs. 311. | |
| Bhatt et al., 2011 [104] | Exclusion criteria: Folstein Mini Mental Status Examination score < 25 or classified as osteopenic or osteoporotic. Definition of fall: if the force recorded on the safety harness load cell force exceeded 30% of the body weight. | Community-dwelling elderly (44 males): 59 fallers (the force recorded on the safety harness load cell force exceeded 30% of the body weight; 71.6 ± 4.6 years). 56 recoveries (71.4 ± 5.1 years). | Subjects walked for 9–12 trials on a 7 m walkway using their own athletic shoes. Kinematic and ground reaction force data were collected during 5 trials using an 8-camera optoelectronic motion analysis system (120 Hz) and one force platform (600 Hz). | | Absolute CoM velocity (m/s): 0.95 vs. 1.0; step length (m): 0.34 vs. 0.34. |
| Kirkwood et al., 2011 [47] | Exclusion criteria: orthopedic or neurological diseases that could affect gait performance. Inclusion criteria: female; ≥60 years; and ability to walk without assistance. Definition of fall: unexpected event in which a subject comes to rest on a lower level. | Community-dwelling elderly women: 45 fallers (≥2 falls in past 12 months; 74.0 ± 5.6 years). 44 non-fallers (absence of falls or 1 fall in past 12 months; 70.7 ± 5.4 years). | Subjects walked barefoot on a 6 m rubber mat while EMG recorded soleus, tibialis anterior, and gastrocnemius muscle signals. Footswitches tracked gait events. | Gastrocnemius activity during stance phase (%): 16.9 vs. 19.8; stride time (s): 1.2 vs. 1.3. | Tibialis anterior activity during stance phase (%): 9.2 vs. 9.3; soleus activity during stance phase (%): 22.1 vs. 24.9; gastrocnemius latency activity (s): 0.30 vs. 0.30; tibialis anterior latency activity (s): 0.04 vs. 0.03; soleus latency activity (s): 0.30 vs. 0.30; swing phase (%): 40.2 vs. 39.7; stance phase (%): 59.8 vs. 60.3. |

**Table 1.** *Cont.*

| Study | Inclusion and/or Exclusion Criteria Definition of Fall | Sample Characteristics | Gait Assessment | Gait Parameters Related to Falls (Fallers vs. Non-Fallers) | Gait Parameters Not Related to Falls (Fallers vs. Non-Fallers) |
|---|---|---|---|---|---|
| Lázaro et al., 2011 [48] | Exclusion criteria: ≥65 years; severe cognitive deterioration; unable to stand; or terminally ill. Inclusion criteria (fallers): had visited their General Practitioner or Geriatrician due to the occurrence of falls. Definition of fall: not reported. | Community-dwelling elderly: 99 fallers (≥2 falls in past 6 months; 17 males; 78.0 ± 5.0 years). 113 non-fallers (no falls in past 6 months). | Gait assessments of subjects were evaluated using the Walk Across test. | Gait speed (m/s): 0.34 vs. 0.50. | |
| Lugade et al., 2011 [49] | Inclusion criteria: no history of head trauma, neurological or heart diseases; muscle, joint, or orthopedic disorder; visual impairment that was uncorrected by glasses; persistent vertigo; or lightheadedness. Definition of fall: not reported. | Community-dwelling elderly: 10 fallers (≥1 fall in past year; 78.9 ± 4.9 years). 10 non-fallers (75.4 ± 7.0 years). | Subjects walked barefoot at preferred gait speed along a 10 m walkway. Kinematic and ground reaction force data were collected during 5 trials using an 8-camera optoelectronic motion analysis system (60 Hz) and two integrated force platforms. CoM and CoP data were calculated. | Gait speed (m/s): 1.02 vs. 1.26; CoM–CoP AP distance at heel strike (cm): 41.6 vs. 52.4. | At heel strike (CoM inside base of support): CoM stability margin (cm): 3.9 vs. 3.5; distance to centroid (cm): 2.5 vs. 2.2; interaction of the CoM position and velocity distance to border (cm): 17.5 vs. 18.7; time to contact (s): 0.17 vs. 0.15; base of support ($m^2$): 0.40 vs. 0.43. At toe-off (CoM outside base of support): CoM separation (cm): 8.3 vs. 10.4; distance to centroid (cm): 21.4 vs. 23.4; time to contact (s): 0.12 vs. 0.11; base of support area ($m^2$): 0.23 vs. 0.22; CoM–CoP ML distance at heel strike (cm): 6.6 vs. 7.3. |
| Panzer et al., 2011 [50] | Exclusion criteria: Mini Mental Status Examination score < 24; body mass index ≥30 kg/$m^2$; blindness; neurologic, orthopedic, or visual disorders that impair mobility; or non-English speaking. Definition of fall: not reported. | Community-dwelling elderly: 47 fallers (≥2 non-injury falls or ≥1 injury fall in past year; 80.1 ± 6.2 years). 27 non-fallers (75.1 ± 6.5 years). | Two self-paced out and back walks (8.1 m) were performed; average gait speed was calculated, and the fastest performance was used. | Average gait speed (m/s): 0.64 vs. 0.86. | |
| Scanaill et al., 2011 [51] | Exclusion and inclusion criteria: not reported. Definition of fall: not reported. | Community-dwelling elderly: 182 fallers (>1 fall in past year or ≥1 fall that resulted in a loss of consciousness, a fractured bone, or severe injury in past year; 40 males; 74.5 ± 7.2 years). 139 non-fallers (60 males; 70.3 ± 6.8 years). | Subjects walked at a preferred gait speed along a 6 m pressure-sensing walkway. Two kinematic sensors were worn on the subject's shanks. | Stride length (m): 1.08 vs. 1.23; stride width (m): 0.12 vs. 0.11; step length (m): 0.54 vs. 0.61; step width (m): 0.56 vs. 0.63. | Stride time (s): 1.23 vs. 1.20; stance time (s): 0.81 vs. 0.79; swing time (s): 0.51 vs. 0.50; step time (s): 0.66 vs. 0.66; single support (%): 75.9 vs. 78.2; double support (%): 34.6 vs. 34.3. Variability (CV): stride length (%): 8.6 vs. 7.8; stride width (%): 25.0 vs. 25.3; step length (%): 14.1 vs. 12.7; step width (%): 12.4 vs. 10.6; base width (%): 24.7 vs. 25.3; stride time (%): 19.2 vs. 18.6; stance time (%): 30.3 vs. 33.0; swing time (%): 32.4 vs. 31.0; step time (%): 34.2 vs. 31.8; single support (%): 21.4 vs. 20.1; double support (%): 61.4 vs. 62.6. |

**Table 1.** *Cont.*

| Study | Inclusion and/or Exclusion Criteria Definition of Fall | Sample Characteristics | Gait Assessment | Gait Parameters Related to Falls (Fallers vs. Non-Fallers) | Gait Parameters Not Related to Falls (Fallers vs. Non-Fallers) |
|---|---|---|---|---|---|
| Uemura et al., 2012 [52] | Exclusion criteria: severe cardiac, pulmonary, or musculoskeletal disorders; diseases associated with a high risk of falling; inability to execute arithmetic tasks; serious visual impairment not correctable with spectacles; or inability to follow multiple commands. Inclusion criteria: ≥65 years; minimal hearing and visual impairments; and ability to ambulate independently. Definition of fall: an event where a subject unintentionally comes to rest on the ground or another lower level; falls resulting from extraordinary environmental factors were excluded. | Community-dwelling elderly (65–93 years): 22 fallers (≥1 fall in past year). 35 non-fallers. | Subjects walked along a 2 m walkway as quickly as possible after a visual cue. CoP data were collected by force plate during 3 trials. Step initiation—first ML deviation of CoP towards swing leg. Reaction phase—time from cue to step initiation. Anticipatory postural adjustment phase—time from step initiation to foot-off. | | Reaction phase (s): 0.31 vs. 0.29; anticipatory postural adjustment phase (s): 0.46 vs. 0.44. |
| Chen & Chou, 2013 [53] | Inclusion criteria: walk without an assistive device; no history of neurological or musculoskeletal deficits (e.g., amputation, cerebral vascular accident, significant head trauma, or Parkinson's disease); and no uncorrectable visual impairment, vestibular dysfunction, or dementia. Definition of fall: unexpected event where the subject falls to the ground from an upper level; falls caused by syncope or major intrinsic events were excluded. | Community-dwelling elderly: 15 fallers (≥2 falls in past year; 4 males; 77.7 ± 7.7 years). 15 non-fallers (6 males; 76.2 ± 4.2 years). | Subjects performed the Timed Up and Go test while barefoot. Kinematic and CoM data and ground reaction force were collected during 4 trials using a 10-camera optoelectronic motion analysis system (600 Hz) and one force platform (960 Hz). | Step length (m): 0.42 vs. 0.52; CoM AP velocity at stance-off (lower fallers); AP inclination of CoM–ankle at stance-off (°): −2.4 vs. −6.8; total CoM kinetic energy at swing-off (J): 6.6 vs. 10.4; total CoM kinetic energy at stance-off (J): 20.6 vs. 31.9. | Step width (m): 0.23 vs. 0.21; CoM AP velocity at swing-off; AP inclination of CoM–ankle at swing-off (°): 7.7 vs. 6.9. |
| Chen et al., 2013 [54] | Inclusion criteria: walk without an assistive device; no history of neurological or musculoskeletal deficits (e.g., amputation, cerebral vascular accident, significant head trauma, or Parkinson's disease); and no uncorrectable visual impairment, vestibular dysfunction, or dementia. Definition of fall: unexpected event where the subject falls to the ground from an upper level; falls caused by syncope or major intrinsic events were excluded. | Community-dwelling elderly: 10 fallers (≥2 falls in past year; 2 males; 75.9 ± 4.1 years). 10 non-fallers (3 males; 75.5 ± 3.0 years). | Subjects performed the Timed Up and Go test while barefoot. Kinematic and CoM data and ground reaction force were collected during 4 trials using a 10-camera optoelectronic motion analysis system (600 Hz) and one force platform (960 Hz). | Braking force (N/kg): −0.83 vs. −0.43 propulsive force (N/kg): 3.48 vs. 5.04; ankle moment at swing-off: 0.11 vs. −0.03. | Trunk angle (°): 32.9 vs. 31.4; hip moment at swing-off (Nm/kg): 0.45 vs. 0.48; knee moment at swing-off (Nm/kg): 0.42 vs. 0.54. |

**Table 1.** *Cont.*

| Study | Inclusion and/or Exclusion Criteria Definition of Fall | Sample Characteristics | Gait Assessment | Gait Parameters Related to Falls (Fallers vs. Non-Fallers) | Gait Parameters Not Related to Falls (Fallers vs. Non-Fallers) |
|---|---|---|---|---|---|
| Chiu & Chou, 2013 [55] | Inclusion criteria: no current histories of neurological or musculoskeletal deficits that affect walking and no uncorrectable visual impairment, vestibular dysfunction, dementia, or depression. Definition of fall: not reported. | Community-dwelling elderly: 15 fallers (≥2 falls in past year; 3 males; 72.9 ± 4.1 years). 15 non-fallers (8 males; 75.7 ± 4.7 years). | Subjects walked barefoot along a 10 m walkway at preferred gait speed. Kinematic data were collected during 5 trials using a 10-camera optoelectronic motion analysis system (60 Hz). SD is used to analyze variability. | Gait speed (m/s): 1.07 vs. 1.22; stance phase (%): 62.6 vs. 60.9; swing phase (%): 37.4 vs. 39.1; single support (%): 37.4 vs. 39.0; double support (%): 25.2 vs. 21.9. Variability in inter-joint coordination during stance phase (SD): knee–ankle (higher fallers); ankle (higher fallers). Variability inter-joint coordination during swing phase (SD): knee–ankle (higher fallers). | Cadence (steps/min): 115 vs. 116. Variability inter-joint coordination during stance phase (SD): hip; knee; hip–knee. Variability inter-joint coordination during swing phase (SD): hip; knee; ankle; hip–knee. |
| Fritz et al., 2013 [36] | Exclusion criteria: orthopedic or neurologic conditions that altered walking. Inclusion criteria: capable of walking unassisted for more than 10 feet and understanding the study's objective. Definition of fall: not reported. | Community-dwelling elderly: 12 fallers (≥1 fall in past 6 months; 86.3 ± 4.7 years). 50 non-fallers (85.4 ± 7.1 years). | Subjects walked during 3 trials at a preferred gait speed along a 6 m pressure-sensing walkway. | Gait speed (m/s): 0.89 vs. 1.0; stride length (m): 0.85 vs. 1.02. | Base of support (cm): 12.3 vs. 10.2; swing phase (%): 31.5 vs. 33.4; stance phase (%): 68.4 vs. 66.6; double support (%): 37 vs. 33; step time variability (CV): 7.3 vs. 6.4. |
| Weiss et al., 2013 [57] | Exclusion criteria: previously clinically diagnosed with any gait or balance disorders and Mini Mental Status Examination score < 24. Definition of fall: any stability disturbance that caused significant contact with the floor. | Community-dwelling elderly: 32 fallers (≥2 fall in past year; 35% males; 77.9 ± 5.1 years). 39 non-fallers (<2 fall in past year; 36% males; 78.8 ± 4.4 years). | Subjects walked for 1 min at preferred gait speed (laboratory assessments). A portable tri-axial accelerometer sensor (100 Hz) was worn on the lower back. Subjects also wore a portable accelerometer sensor (100 Hz) for 3 days. | Laboratory assessment: gait speed (m/s): 0.97 vs. 1.19; step duration (s): 0.55 vs. 0.52. 3-day assessment: step time; stride time. Fallers presented higher variability in the lower back vertical axis and lower variability in the lower back ML axis. | |
| Marques et al., 2013 [59] Marques et al., 2013 [58] | Exclusion criteria: Mini Mental Status Examination score < 20; cardiovascular disease; Berg balance scale score < 36; hemiparesis; pain of the lower limbs or trunk; or progressive motor disorder. Definition of fall: any stability disturbance that caused significant contact with the floor. | Community-dwelling elderly women: 15 fallers (≥1 fall in past year; 69.6 ± 8.0 years). 22 non-fallers (66.1 ± 6.2 years). | Subjects walked at preferred gait speed for 1 min on a walkway and for 10 min on a treadmill. Gait kinematic parameters and EMG activity were assessed using a 7-camera optoelectronic motion analysis system (100 Hz) and an 8-channel telemetry EMG system (2000 Hz). | Hip position at toe-off (°): 9.5 vs. 5.4; muscle activation at initial stance: biceps femoris (%): 36.4 vs. 24.1; muscle activation at final stance: gluteus maximus (%): 86.4 vs. 52.3; muscle activation before heel contact: internal oblique (%): 8.3 vs. 15.7; biceps femoris (%): 45.5 vs. 31.3. | Gait speed on walkway (m/s): 1.1 vs. 1.3; gait speed on treadmill (m/s): 0.9 vs. 0.9; step time (s): 0.23 vs. 0.26; step length (m): 0.51 vs. 0.50; step width (m): 0.14 vs. 0.17; ankle angular position at heel contact (°): 6.4 vs. 5.9. Muscle activation at initial stance: internal oblique (%): 97.2 vs. 100.3; rectus femoris (%): 143.8 vs. 130.6; tibialis anterior (%): 106.7 vs. 122.8; multifidus (%): 150.5 vs. 147.7, gluteus maximus (%): 154.7 vs. 179.9. Muscle activation at final stance: internal oblique (%): 117.5 vs. 105.1; rectus femoris (%): 89.5 vs. 80.1, |

**Table 1.** *Cont.*

| Study | Inclusion and/or Exclusion Criteria Definition of Fall | Sample Characteristics | Gait Assessment | Gait Parameters Related to Falls (Fallers vs. Non-Fallers) | Gait Parameters Not Related to Falls (Fallers vs. Non-Fallers) |
|---|---|---|---|---|---|
| | | | | | multifidus (%): 76.1 vs. 82.8; biceps femoris (%): 43.8 vs. 50.1; gastrocnemius lateralis (%): 91.7 vs. 75.8. Muscle activation before heel contact: rectus femoris (%): 12.7 vs. 15.2; tibialis anterior (%): 40.0 vs. 30.1; multifidus (%): 16.4 vs. 18.4; gluteus maximus (%): 12.0 vs. 16.8; gastrocnemius lateralis (%): 7.2 vs. 14.9. Muscle activation after toe-off: internal oblique (%): 21.6 vs. 20.5; rectus femoris (%): 10.9 vs. 15.9; tibialis anterior (%): 35.1 vs. 31.9; gluteus maximus (%): 6.8 vs. 10.1; biceps femoris (%): 16.2 vs. 13.1; gastrocnemius lateralis (%): 7.9 vs. 12.2. |
| Ayoubi et al., 2014 [60] | Exclusion criteria: <65 years; institutionalization; non–French-speaking; acute medical illness during the past month; diagnosis of dementia; score > 2 on item 22 of Unified Parkinson's Disease Rating Scale; severe orthopedic diagnoses of lumbar vertebra, pelvis, or lower extremities; or inability to walk 6 m unassisted. Definition of fall: subject unintentionally coming to rest on the ground or other lower level, and not as the result of a major intrinsic event. | Community-dwelling elderly: 109 fallers with a fear of falling (24 males; 71 ± 5.2 years). 101 fallers with no fear of falling (29 males; 70.8 ± 5.5 years). 194 non-fallers with fear of falling (83 males; 70.5 ± 5.0 years). 619 non-fallers with no fear of falling (368 males; 70.3 ± 4.8 years). | Subjects walked 1 trial at their preferred gait speed along a 6 m pressure-sensing walkway. | Fallers with fear of falling vs. non-fallers with no fear of falling: gait speed (m/s): 0.96 vs. 1.11; stride time variability (CV; %): 3.0 vs. 2.0. | Fallers with no fear of falling vs. non-fallers with no fear of falling: gait speed (m/s) 1.07 vs. 1.11; stride time variability (CV; %): 2.0 vs. 2.0. Fallers with no fear of falling vs. non-fallers with fear of falling: gait speed (m/s) 1.07 vs. 1.03; stride time variability (CV; %): 2.0 vs. 3.0. Fallers with fear of falling vs. non-fallers with fear of falling: gait speed (m/s) 0.96 vs. 1.03; stride time variability (CV; %): 3.0 vs. 3.0. |
| Barelle et al., 2014 [61] | Exclusion criteria: vascular stroke with motor or sensory after-effects; Parkinson's disease; hip or knee prosthesis; or fracture of leg or ankle which would have impaired gait. Definition of fall: not reported. | Community-dwelling elderly: 6 fallers (≥1 fall in past 6 months; 68.0 ± 4.0 years). 6 non-fallers (2 males; 69.0 ± 3.0 years). | Subjects walked at preferred gait speed on a 10 m walkway. Gait kinematic parameters were assessed using an 8-camera optoelectronic motion analysis system (100 Hz). | | Stride and step length (m): 1.13 vs. 1.18; 0.57 vs. 0.59; stride length (% height): 70 vs. 74; cadence (strides/s): 0.87 vs. 0.92; cadence (steps/s): 1.73 vs. 1.84; step length (% height): 35 vs. 37; cycle time (s): 1.17 vs. 1.09; gait speed (m/s): 0.99 vs. 1.08; hip, knee, ankle displacements (°): 21 vs. 21; 58 vs. 58; 38 vs. 37. |

**Table 1.** *Cont.*

| Study | Inclusion and/or Exclusion Criteria Definition of Fall | Sample Characteristics | Gait Assessment | Gait Parameters Related to Falls (Fallers vs. Non-Fallers) | Gait Parameters Not Related to Falls (Fallers vs. Non-Fallers) |
|---|---|---|---|---|---|
| Iwata et al., 2014 [62] | Exclusion and inclusion criteria: not reported. Definition of fall: any unintended contact with a supporting surface. | Community-dwelling elderly: 28 fallers ($\geq$1 fall in past year; 9 males; 76.0 $\pm$ 5.3 years). 84 non-fallers (19 males; 73.5 $\pm$ 6.1 years). | Maximum gait speed was measured using a floor-based photocell gait analysis system over a 5 m course. | | Maximum gait speed (m/s): 1.8 vs. 1.9. |
| Kobayashi et al., 2014 [63] | Inclusion criteria: walk independently; normal or corrected-to-normal vision; and no history of neuromuscular disease. Definition of fall: not reported. | Community-dwelling elderly: 18 fallers ($\geq$1 fall in past year; 67.3 $\pm$ 3.1 years). 19 non-fallers (67.1 $\pm$ 3.3 years). | Subjects walked at preferred gait speed on a 10 m walkway, during 5 trials. Gait kinematic and ground reaction force data were assessed using an optoelectronic motion analysis system (200 Hz) and six force platforms (1000 Hz). Principal component analysis was used to analyze the relationship between the risk of falling and the joint kinematics of the lower limbs. MTC was analyzed. | Gait speed (m/s): 1.21 vs. 1.33; stance time variability (SD; s): 0.014 vs. 0.009. Fallers exhibited greater variability in the hip, knee, and ankle in all planes during the entire swing phase. Fallers exhibited greater variability in the hip and ankle in the frontal plane during the entire stance phase. Fallers exhibited smaller hip flexion and ankle dorsiflexion angles between the mid-stance and late-stance phases. Fallers exhibited larger ankle inversion between the mid-stance and late-stance phases. Fallers exhibited smaller hip abduction during the mid-stance phase. Variability in the joint kinematics is the key characteristic that affects the risk of falling while walking. | MTC (cm): 4.30 vs. 4.24; step length (m): 0.63 vs. 0.67; step width (m): 0.10 vs. 0.09; stance time (s): 0.60 vs. 0.58; swing time (s): 0.41 vs. 0.41; stance phase (%): 64.8 vs. 63.9; MTC variability (SD; cm): 0.27 vs. 0.29; gait speed variability (SD; m/s): 0.04 vs. 0.03; step length variability (SD; m): 17.8 vs. 13.5; step width variability (SD; cm): 16.3 vs. 16.8; swing time variability (SD; s): 0.02 vs. 0.01; stance phase variability (SD; %): 0.90 vs. 0.76. |
| König et al., 2014 [64] | Exclusion criteria: body mass index < 18 or >33 kg/m$^2$; alcoholism; type-1 diabetes; cardiac infarct; chronic hepatitis; celiac and malabsorption diseases; rheumatoid arthritis; cancer; treated for more than 3 months or under treatment with oral corticosteroids; hyperparathyroidism; hyperthyroidism; neurological diseases affecting neuromuscular system; peripheral neurologic diseases; fractures or osteosyntheses; total hip replacement (less than 6 months); | Community-dwelling elderly: 38 fallers ($\geq$1 fall in past year; 69.2 $\pm$ 4.8 years). 42 non-fallers (68.9 $\pm$ 4.5 years). | Subjects walked barefoot at preferred gait speed on a 10 m walkway during 6 trials. Gait kinematic parameters were assessed using an optoelectronic motion analysis system (200 Hz). Principal component analysis was used. | Temporal variability and mean spatial gait parameters. | |

**Table 1.** *Cont.*

| Study | Inclusion and/or Exclusion Criteria Definition of Fall | Sample Characteristics | Gait Assessment | Gait Parameters Related to Falls (Fallers vs. Non-Fallers) | Gait Parameters Not Related to Falls (Fallers vs. Non-Fallers) |
|---|---|---|---|---|---|
| | unable to follow instructions or unable to walk 10 m without a walking aid; or participation in another study at the same time. Definition of fall: not reported. | | | | |
| Mignardot et al., 2014 [65] | Exclusion criteria: refusal or lack of capacity to give consent or hospitalized at the time of screening. Inclusion criteria: 66–75 years; living at home; never fallen; and ability to walk without assistance for at least 30 s. Definition of fall: unintentionally fall on the ground or lower level, not as a result of a major intrinsic event (e.g., as a stroke) or overwhelming hazard. | Community-dwelling elderly: 72 fallers (≥1 fall in 2 years follow-up; 35 males; 71.1 ± 2.7 years). 187 non-fallers (72 males; 69.4 ± 2.5 years). | Subjects walked with their own shoes at preferred gait speed along a 30 m walkway. A tri-axial accelerometer sensor was used (100 Hz). Principal component analysis was used to assess the relationship between gait variables and fall status. | PC1—global kinetics of gait pattern (mechanical power and spatiotemporal variables): fallers (+0 to +6 months) differed from non-fallers and fallers (+6 to +12 months); PC1 had predictive power for the first fall onset during the first six months after the initial screening. PC2—global gait regularity: fallers (+6 to +12 months) differed from non-fallers (+0 to +6 months); PC2 had predictive power for the first fall onset between the 6th and 12th months after initial screening. | PC3—stride time: there was no significant difference between fallers and non-fallers on PC3; PC3 did not have any predictive power for the first fall onset. |
| Cebolla et al., 2015 [66] | Inclusion criteria: ≥60 years; able to perform activities of daily living and walk independently; and no orthopedic problems (e.g., surgery or fractures) or other health problems that impair physical tests. Definition of fall: unintentionally coming to rest on the ground or other lower level, whether or not it produced an injury. | Community-dwelling elderly (13 males): 20 fallers (≥1 fall in past year; 68.0 ± 6.9 years). 42 non-fallers (65.5 ± 4.1 years). | Subjects walked at preferred gait speed on an 8 m walkway during 10 trials. Gait kinematic parameters were assessed using a 6-camera optoelectronic motion analysis system (100 Hz). MTC was analyzed. | MTC (mm): 40 vs. 43. | Stride length (m): 1.11 vs. 1.17; stride time (s): 1.07 vs. 1.08; cadence (strides/s): 0.93 vs. 0.93; gait speed (m/s): 1.04 vs. 1.08; heel vertical velocity at heel strike (m/s): 0.70 vs. 0.76. |
| MacAulay et al., 2015 [67] | Exclusion criteria: Geriatric Depression Scale score ≥ 6 or neurologic or untreated health disorders (e.g., cerebrovascular disease, Parkinson's disease, traumatic brain injury). Definition of fall: subject unexpectedly lost his stability and unintentionally came unto rest on the ground or other object; events in which participants were able to regain their stability did not count as a fall. | Community-dwelling elderly (128 males): 81 fallers (≥1 fall in past year; 69.9 ± 6.8 years). 312 non-fallers (70.1 ± 6.6 years). | Subjects walked at preferred gait speed along a 6 m pressure-sensing walkway. Four trials were collected. | Stride length (lower fallers). | Step time (s): 1.08 vs. 1.08. |

**Table 1.** *Cont.*

| Study | Inclusion and/or Exclusion Criteria Definition of Fall | Sample Characteristics | Gait Assessment | Gait Parameters Related to Falls (Fallers vs. Non-Fallers) | Gait Parameters Not Related to Falls (Fallers vs. Non-Fallers) |
|---|---|---|---|---|---|
| Rispens et al., 2015 [68] | Inclusion criteria: Mini Mental Status Examination score > 18 and able to walk at least 20 m with a walking aid. Definition of fall: event that resulted in unintentionally coming to rest on the ground or other lower level. | Community-dwelling elderly (33 males; 78.4 ± 7.8 years): 41 fallers (≥1 fall in past year). 69 non-fallers. | Subjects wear a portable tri-axial accelerometer sensor (100 Hz) for 2 separate weeks. It was attached with an elastic belt around the waist and set along the lumbar spine. Subjects were instructed to wear the accelerometer at all times, except during water activities. Intra-class correlation was used. | Gait speed; gait speed variability (SD); stride time; stride time variability (SD); gait symmetry (harmonic ratio); and gait smoothness (index of harmonicity) were associated with the number of falls in the past year. | Cadence variability (SD). |
| Wright et al., 2015 [69] | Inclusion criteria: able to walk at least 100 m without the use of a gait aid and no neurological disease, head trauma, musculoskeletal impairment, or visual impairment not correctable by lenses. Definition of fall: a loss of balance resulting in the body, or part of the body, coming to rest on the ground. | Community-dwelling elderly: 14 "trip" fallers (≥1 trip fall in past year; 4 males; 71 ± 6 years). 10 "slip" fallers (≥1 slip fall in past year; 4 males; 68 ± 5 years). 16 non-fallers (6 males; 72 ± 5 years). | Subjects walked at preferred gait speed along a walkway. Kinematic and ground reaction forces data were collected during 3 trials using a 14-camera optoelectronic motion analysis system (60 Hz) and two force platforms (120 Hz). CoM and CoP data were calculated. | Differences between fallers (both groups) and non-fallers: CoM–CoP at heel strike (cm): 14.3 vs. 15.3 vs. 12.0. Differences between "slip" fallers and non-fallers: CoM–CoP at foot flat (cm): −14.9 vs. −10.3. Differences between "trip" fallers and "slip" fallers: CoM–CoP at mid-swing (cm): 0.9 vs. 1.2. | "Trip" fallers vs. "slip" fallers vs. non-fallers: gait speed (m/s): 1.19 vs. 1.22 vs. 1.14; stride time (s): 1.06 vs. 1.10 vs. 1.10; stride length (m): 1.26 vs. 1.34 vs. 1.26; CoM–CoP at toe-off (cm): −15.1 vs. −16.5 vs. −14.3; CoM–CoP at late swing (cm): 13.4 vs. 13.2 vs. 11.0; peak braking force (% body mass): −15.9 vs. −16.5 vs. −15.1; instant of peak braking force (% gait cycle): 10.8 vs. 11.2 vs. 10.9; peak propulsive force (% body mass): 17.3 vs. 19.3 vs. 17.1; instant of peak propulsive force (% gait cycle): 54.0 vs. 54.1 vs. 54.3. |
| Bounyong et al., 2016 [70] | Exclusion and inclusion criteria: not reported. Definition of fall: not reported. | Community-dwelling elderly (8 males; 72.3 ± 6.1 years): 17 fallers (≥1 fall in past year). 35 non-fallers. | Subjects walked 6 trials at preferred gait speed along a 5 m walkway. EMG of rectus femoris, biceps femoris, tibialis anterior, and gastrocnemius were collected (1024 Hz). Co-contraction index was determined based on EMG. | Co-contraction index (between tibialis anterior and gastrocnemius) during stance phase (%): 61.8 vs. 57.5. | |
| Fujimoto & Chou, 2016 [71] | Inclusion criteria: no history or clinical evidence of neurological, musculoskeletal, or other medical conditions (neurological pathology, head trauma, cerebrovascular accident, vestibular dysfunction, or visual impairment uncorrectable by lenses). Definition of fall: not reported. | Community-dwelling elderly: 15 fallers (≥2 falls in past year; 3 males; 71.9 ± 4.3 years). 15 non-fallers (6 males; 70.0 ± 3.2 years). | Subjects walked barefoot at preferred gait speed along a 10 m walkway. Kinematic data were collected during 6 trials using an 8-camera optoelectronic motion analysis system (60 Hz). CoM was calculated. | CoM position at toe-off (m/s): −0.30 vs. −0.47; CoM mean velocity (m/s): 1.03 vs. 1.29; CoM mean velocity at toe-off (m/s): 1.29 vs. 1.61; CoM acceleration peak prior to toe-off (m/s): 0.38 vs. 0.49; CoM AP acceleration peak (lower fallers). | |

**Table 1.** *Cont.*

| Study | Inclusion and/or Exclusion Criteria Definition of Fall | Sample Characteristics | Gait Assessment | Gait Parameters Related to Falls (Fallers vs. Non-Fallers) | Gait Parameters Not Related to Falls (Fallers vs. Non-Fallers) |
|---|---|---|---|---|---|
| Ihlen et al., 2016 [72] | Exclusion and inclusion criteria: not reported. Definition of fall: not reported. | Community-dwelling elderly (78.4 ± 4.7 years): 32 fallers (≥2 falls in past year). 39 non-fallers. | Subjects wear a tri-axial accelerometer (100 Hz) for 3 days over the lower back. The refined composite multiscale entropy and refined multiscale permutation entropy were applied to trunk acceleration and velocity signals in the AP, ML, and vertical directions. | Refined composite multiscale entropy is higher for non-fallers compared to fallers for trunk AP, ML, and vertical acceleration. Refined multiscale permutation entropy is higher for non-fallers compared to fallers for trunk ML acceleration in the intermediate and large scales. Refined multiscale permutation entropy is lower for non-fallers compared to fallers for trunk vertical acceleration in the intermediate and large scales. | |
| Howcroft et al., 2016 [73] | Exclusion criteria: cognitive disorder (self-reported) or unable to walk for 6 min without an assistive device. Inclusion criteria: ≥65 years. Definition of fall: event that results in a subject coming to rest unintentionally on the ground or other lower level, excluding falls resulting from a stroke or overwhelming hazard. | Community-dwelling elderly: 24 fallers (≥1 fall in past 6 months; 13 males; 76.3 ± 7.0 years). 76 non-fallers (31 males; 75.3 ± 6.6 years). | Subjects walked 7.62 m while wearing pressure-sensing insoles (120 Hz) and tri-axial accelerometers on the head, pelvis, and left and right shanks (50 Hz). CoP data were analyzed. Maximum Lyapunov exponent, ratio of even to odd harmonics, SD, and CV are used to analyze data variability. | Head variability (SD; higher fallers); ratio of even to odd harmonics pelvis AP (lower fallers). | CoP ML deviation time (s): 0.03 vs. 0.03; minimum CoP velocity (m/s): 0.03 vs. 0.03; mean CoP velocity (m/s): 0.30 vs. 0.30; median CoP velocity (m/s): 0.24 vs. 0.21; gait speed (m/s): 1.24 vs. 1.20; cadence (steps/min): 112 vs. 111; stride time (s): 1.09 vs. 1.09; stance time (s): 0.71 vs. 0.72; swing time (s): 0.38 vs. 0.38; stride time (CV; %): 3 vs. 3; stance time (CV; %): 5 vs. 6; swing time (CV; %): 8 vs. 11; stance phase (%): 64.6 vs. 65.9; double-support phase (%): 14.6 vs. 15.9; CoP AP displacement (CV; %): 495 vs. 463; CoP ML displacement (CV; %): 650 vs. 666; Impulse during foot-strike to first peak (Ns/kg): 1.20 vs. 1.20; Impulse during MTC to second peak (Ns/kg): 1.47 vs. 1.67; Impulse during second peak to foot-off (Ns/kg): 0.97 vs. 1.08; Impulse during foot-strike to MTC (Ns/kg): 2.35 vs. 2.40; Impulse during MTC to foot-off (Ns/kg): 2.36 vs. 2.67; Impulse during foot-strike to foot-off (Ns/kg): 4.65 vs. 4.99; maximum Lyapunov exponent. |

**Table 1.** *Cont.*

| Study | Inclusion and/or Exclusion Criteria Definition of Fall | Sample Characteristics | Gait Assessment | Gait Parameters Related to Falls (Fallers vs. Non-Fallers) | Gait Parameters Not Related to Falls (Fallers vs. Non-Fallers) |
|---|---|---|---|---|---|
| Rinaldi & Moraes, 2016 [74] Rinaldi et al., 2017 [75] | Exclusion criteria: Mini Mental Status Examination score < 24; vestibular dysfunction; or unable to walk without assistance. Inclusion criteria: no history of neurological or musculoskeletal disorders and no incorrigible visual impairment. Definition of fall: event in which a subject comes unintentionally to the ground or to some lower level. | Community-dwelling elderly women: 15 fallers (≥1 fall in past year; 70.1 ± 5.1 years). 15 non-fallers (71.8 ± 5.8 years). | Subjects walked at preferred gait speed. Gait kinematic data were assessed using an 8-camera optoelectronic motion analysis system (100 Hz). CoM data were analyzed. | Gait speed (m/s): 1.06 vs. 1.23; step width (m): 0.09 vs. 0.06; step time (s): 0.62 vs. 0.52; gait speed (m/s): 0.64 vs. 0.93; CoM AP velocity (m/s): 0.39 vs. 0.75; percentage of CoM AP velocity (%): 60 vs. 30; margin of dynamic stability in AP direction (m): 0.07 vs. 0.02; margin of dynamic stability in ML direction (m): 0.04 vs. 0.01. | |
| Bizovska et al., 2017 [76] | Exclusion criteria: musculoskeletal problems or injuries and surgical interventions that were performed within 2 years of the study. Inclusion criteria: ≥60 years and ability to stand and walk without any support. Definition of fall: unexpected event in which the participants come to rest on the ground or lower level. Falls related to sports, such as skiing and cycling, and those caused by a great external force were excluded. | Community-dwelling elderly: 38 fallers (≥1 fall in 6 months follow-up; median 70.9 years). 101 non-fallers (median 70.6 years). | Subjects walked at preferred gait speed along a 30 m walkway for 5 min wearing comfortable sports shoes. Tri-axial accelerometers were attached to L5 and shanks (296.3 Hz). The index of complexity, the computed from multiscale entropy, and the Shannon entropy were used to analyze data variability. | ShE: trunk AP direction: 0.34 vs. 0.31; ShE shanks ML direction: 0.41 vs. 0.37. | Gait speed (m/s): 1.22 vs. 1.23; stride time (s): 1.03 vs. 1.05. Shannon entropy: trunk vertical direction: 0.44 vs. 0.43; trunk ML direction: 0.16 vs. 0.17; shanks vertical direction: 0.59 vs. 0.57; shanks AP direction: 0.58 vs. 0.58. Index of complexity: trunk vertical direction: 12.5 vs. 12.4; trunk ML direction: 17.3 vs. 18.0; trunk AP direction: 9.9 vs. 10.2; shanks vertical direction: 9.0 vs. 8.6; shanks ML direction: 15.20 vs. 15.20; shanks AP direction: 8.5 vs. 8.5. Computed from multiscale entropy. |
| de Melker Worms et al., 2017 [77] | Exclusion criteria: Mini Mental Status Examination score < 25; rheumatoid arthritis in lower extremities; cerebral vascular disease; Parkinson's disease; peripheral neuropathy; cardiac arrest; bypass treatment; any other neurological or cardiovascular impairment; or unable to walk for 10 min without a walking aid. Definition of fall: event in which a subject unintentionally comes to rest on the ground or other lower level. | Community-dwelling elderly (8 males): 9 fallers (≥1 fall in past year; 70.4 ± 3.6 years). 19 non-fallers (69.3 ± 3.6 years). | Subjects walked at 1 m/s on a treadmill. Two bouts of 5 min of walking. Gait kinematic data were assessed using a 10-camera optoelectronic motion analysis system. CoM was analyzed. | Stance time (CV; %): 3.5 vs. 3.0; local divergence exponent of the CoM velocity: 0.97 vs. 0.88. | Step length (m): 0.51 vs. 0.55; step width (m): 0.15 vs. 0.13; stance time (s): 0.69 vs. 0.73; swing time (s): 0.38 vs. 0.41; step length (CV; %): 4.5 vs. 4.2; step width (CV; %): 15.6 vs. 18.6; swing time (CV; %): 4.9 vs. 4.4. |

**Table 1.** *Cont.*

| Study | Inclusion and/or Exclusion Criteria Definition of Fall | Sample Characteristics | Gait Assessment | Gait Parameters Related to Falls (Fallers vs. Non-Fallers) | Gait Parameters Not Related to Falls (Fallers vs. Non-Fallers) |
|---|---|---|---|---|---|
| de Melker Worms et al., 2017 [78] | Exclusion criteria: Mini Mental State Examination score < 25. Inclusion criteria: ≥65 years and able to walk independently for 10 min. Definition of fall: event in which a subject unintentionally comes to rest on the ground or other lower level. | Community-dwelling elderly: 8 fallers (≥1 fall in past year). 17 non-fallers. | Subjects walked at 1 m/s on a treadmill. Two bouts of 5 min of walking and two slips were induced. Gait kinematic data were assessed using a 10-camera optoelectronic motion analysis system (100 Hz). | | Step length of the recovery step; step width of the recovery step; step length variability in the recovery step (CV); step width variability in the recovery step (CV). |
| Júnior et al., 2017 [80] | Exclusion criteria: unable to walk without help; severe impairment of stability; or Mini Mental Status Examination score < 13 for elderly illiterate, <18 for 1–7 years of education, <26 for ≥8 years of education. Definition of fall: event in which a subject comes to rest on the ground or lower level. | Community-dwelling elderly: 27 fallers (1 fall in past 6 months; 1 male; 68.0 ± 5.7 years). 35 non-fallers (11 males; 68.0 ± 4.8 years). | Subjects walked at preferred gait speed during 3 trials along an 8 m pressure-sensing walkway. | | Gait speed (m/s): 1.12 vs. 1.27 (statistical tendency for difference, $p$ = 0.060); cadence (steps/min): 113 vs. 112; step length (m): 0.60 vs. 0.63; stride time (s): 1.06 vs. 1.07; single support phase (%): 37.6 vs. 38.4; stride time variability (CV; %): 2.8 vs. 2.7. |
| Marques et al., 2017 [79] | Exclusion criteria: musculoskeletal pain, fractures, or severe soft tissue injury during the previous 6 months or neurological, cardiovascular, or respiratory diseases. Definition of fall: any stability disturbance that caused a subject's body to have significant contact with the floor. | Community-dwelling elderly women: 16 fallers (≥1 injury fall in past year; 69.6 ± 8.1 years). 19 non-fallers (66.1 ± 6.2 years). | Subjects walked on a treadmill at preferred gait speed. Kinematic data were collected using a telemetry data acquisition system and gait phases using pressure sensors (2000 Hz). SDNN: SD of all time intervals. SDANN: SD of means of intervals taken every five strides. SDNNi: mean of SD of intervals. rMSSD: root-mean-square of differences between intervals. Triangular index: geometric method calculated based on a histogram of intervals. | Stance time: SDNN (higher fallers); SDNNi (higher fallers); rMSSD (higher fallers); CV (higher fallers). Swing time: SDANN (higher fallers). Step time: SDNN (higher fallers); SDNNi (higher fallers); rMSSD (higher fallers); triangular index (higher fallers). | Preferred gait speed (m/s): 0.90 vs. 0.90. Stance time: SDANN; triangular index. Swing time: SDNN; DNNi; rMSSD; CV; triangular index. Step time: SDANN; CV. |
| Svoboda et al., 2017 [81] | Exclusion criteria: neurological or vestibular diseases or surgery in lower limbs or spine in the last 2 years. Inclusion criteria: ≥60 years; ability to walk without an assistive device; and ability to stand unassisted without any support during common everyday activities. Definition of fall: unexpected event in which the subject comes to rest on the ground or lower level. | Community-dwelling elderly: 31 fallers (≥1 fall in past 6 months; 4 males; 70.9 ± 6.2 years). 94 non-fallers (19 males; 70.4 ± 6.6 years). | Subjects walked barefoot on a 10 m walkway. Each subject participated in 5 trials at preferred, defined (between 1.00 and 1.22 m/s), and fast gait speed. Ground reaction force data were collected using two force platforms. CV and SD were used to analyze variability. CoP ML and AP displacements were calculated. | Preferred gait speed: gait speed (CV; %): 5.9 vs. 5.0. Defined gait speed: gait speed (CV; %): 6.1 vs. 5.0; CoP ML displacement variability during pre-swing (SD; mm): 1.14 vs. 0.85. Fast gait speed: step width variability (CV; %): 27.7 vs. 22.7. | Preferred gait speed: step length: 0.58 vs. 0.59; step width (cm): 9.5 vs. 10.3; step time (s): 0.53 vs. 0.53; gait speed (m/s): 1.11 vs. 1.13; step length (CV; %): 3.1 vs. 3.1; step width (CV; %): 23.7 vs. 24.3; step time (CV; %): 4.1 vs. 3.5; CoP ML and AP displacements variabilities during loading response (mm): 3.11 vs. 3.20; 5.36 vs. 5.03; CoP ML and AP |

**Table 1.** *Cont.*

| Study | Inclusion and/or Exclusion Criteria Definition of Fall | Sample Characteristics | Gait Assessment | Gait Parameters Related to Falls (Fallers vs. Non-Fallers) | Gait Parameters Not Related to Falls (Fallers vs. Non-Fallers) |
|---|---|---|---|---|---|
| | | | | | displacements variabilities during mid-stance (mm): 0.16 vs. 0.16; 0.45 vs. 0.47; CoP ML and AP displacements variabilities during terminal stance (mm): 0.15 vs. 0.15; 0.56 vs. 0.58; CoP ML and AP displacements variabilities during pre-swing (mm): 0.99 vs. 0.87; 3.16 vs. 2.37. Defined gait speed: step length: 0.58 vs. 0.59; step width (cm): 9.7 vs. 10.0; step time (s): 0.52 vs. 0.53; gait speed (m/s): 1.12 vs. 1.11; step length (CV; %): 3.3 vs. 3.0; step width (CV; %): 24.6 vs. 26.3; step time (CV; %): 4.1 vs. 3.7; COP ML and AP displacements variabilities during loading response (mm): 3.11 vs. 3.10; 6.03 vs. 5.18; CoP ML and AP displacements variabilities during mid-stance (mm): 0.15 vs. 0.16; 0.43 vs. 0.44; CoP ML and AP displacements variabilities during terminal stance (mm): 0.15 vs. 0.15; 0.57 vs. 0.57; CoP AP displacement variability during pre-swing (mm): 3.59 vs. 2.22. Fast gait speed: step length: 0.65 vs. 0.66; step width (cm): 9.6 vs. 10.4; step time (s): 0.43 vs. 0.44; gait speed (m/s): 1.53 vs. 1.50; step length (CV; %): 3.7 vs. 3.3; step time (CV; %): 3.8 vs. 3.6; gait speed (CV; %): 5.2 vs. 4.6; CoP ML and AP displacements variabilities during loading response (mm): 4.44 vs. 3.91; 8.54 vs. 8.06; CoP ML and AP displacements variabilities during mid-stance (mm): 0.25 vs. 0.23; 1.06 vs. 0.97; CoP ML and AP displacements variabilities during terminal stance (mm): 0.17 vs. 0.17; 0.81 vs. 0.69. |

**Table 1.** *Cont.*

| Study | Inclusion and/or Exclusion Criteria Definition of Fall | Sample Characteristics | Gait Assessment | Gait Parameters Related to Falls (Fallers vs. Non-Fallers) | Gait Parameters Not Related to Falls (Fallers vs. Non-Fallers) |
|---|---|---|---|---|---|
| Allen & Franz, 2018 [82] | Exclusion criteria: body mass index $\geq$ 30 kg/m$^2$; sedentary lifestyle; neurologic or orthopedic diseases; taking medication that causes dizziness; or normal or corrected to normal vision. Definition of fall: unintentionally coming to the ground or some lower level, and other than as a sustaining violent blow, loss of consciousness, or sudden onset of paralysis. | Community-dwelling elderly: 10 fallers ($\geq$1 fall in past year; 3 males; 77.7 $\pm$ 7.7 years). 11 non-fallers (5 males; 75.1 $\pm$ 5.8 years). | Preferred gait speed on a walkway was calculated using two photocells. Subjects walked at preferred gait speed on a treadmill. Kinematic data were assessed using a 14-camera optoelectronic motion analysis system (100 Hz). EMG of leg was recorded (1000 Hz). | Muscle synergy: 2.7 vs. 3.1. Variance in leg muscle recruitment accounted for by one module (larger fallers). | Preferred gait speed (m/s): 1.04 vs. 1.17 (statistical tendency for the difference, $p$ = 0.060). Hip flexion and adduction angular position peaks; knee flexion and ankle dorsiflexion angular position peaks. |
| Benson et al., 2018 [83] | Exclusion criteria: able to walk without an assistive device for 5 min or Mini Mental State Examination score < 22. Definition of fall: unintentionally coming to rest on the ground. | Community-dwelling elderly: 10 fallers ($\geq$1 fall in past 6 months; 5 males; 75.3 years). 10 non-fallers (3 males; 71.9 years). | Subjects walked at preferred gait speed on a treadmill with laboratory shoes. Kinematic data were assessed using a 10-camera optoelectronic motion analysis system (100 Hz). | | Knee displacement; knee angle peak. |
| Howcroft et al., 2018 [86] | Exclusion criteria: cognitive disorder (self-reported) or unable to walk for 6 min without an assistive device. Inclusion criteria: $\geq$65 years and without a fall in the 6 months before evaluation. Definition of fall: event that results in a subject coming to rest unintentionally on the ground or other lower level, excluding falls resulting from a stroke or overwhelming hazard. | Community-dwelling elderly: 28 fallers ($\geq$1 fall in 6 months follow-up; 14 males; 75.0 $\pm$ 8.2 years). 47 non-fallers (17 males; 75.3 $\pm$ 5.5 years). | Subjects walked 7.62 m while wearing pressure-sensing insoles (120 Hz) and tri-axial accelerometers on the head, pelvis, and left and right shanks (50 Hz). CoP AP and ML displacements were calculated. Fast Fourier transform first quartile and CV were used to analyze data variability. | Fast Fourier transform first quartile of left shank ML displacement was lower in fallers. Fast Fourier transform first quartile of right shank vertical displacement was lower in fallers. | Gait speed (m/s): 1.17 vs. 1.22; cadence (steps/min): 110 vs. 112; stride time (s): 1.11 vs. 1.09; stance time (s): 0.73 vs. 0.72; swing time (s): 0.38 vs. 0.37; stride time (CV; %): 3.0 vs. 3.0; stride time symmetry index: 2.13 vs. 2.18; lateral deviation length (mm): 0.9 vs. 1.0; ML deviation time (s): 0.03 vs. 0.03; CoP minimum velocity (m/s): 0.03 vs. 0.03; CoP mean velocity (m/s): 0.28 vs. 0.29; CoP median velocity (m/s): 0.25 vs. 0.25; CoP AP (CV; %): 4.9 vs. 4.5; CoP ML (CV; %): 6.6 vs. 6.7. Impulse foot-strike to 1st peak (Ns/kg): 1.22 vs. 1.20; Impulse 1st peak to minimum (Ns/kg): 1.22 vs. 1.27; impulse minimum to 2nd peak (Ns/kg): 1.83 vs. 1.58; impulse 2nd peak to foot-off (Ns/kg): 1.14 vs. 1.05; impulse foot-strike to minimum (Ns/kg): 2.36 vs. 2.44; impulse minimum to foot-off (Ns/kg): 2.89 vs. 2.56; impulse foot-strike to foot-off (Ns/kg): 5.19 vs. 4.89. |

**Table 1.** *Cont.*

| Study | Inclusion and/or Exclusion Criteria Definition of Fall | Sample Characteristics | Gait Assessment | Gait Parameters Related to Falls (Fallers vs. Non-Fallers) | Gait Parameters Not Related to Falls (Fallers vs. Non-Fallers) |
|---|---|---|---|---|---|
| Kwon et al., 2018 [84] | Exclusion criteria: not walking independently without assistance devices or any diseases that affected physical activity (e.g., musculoskeletal disease, neurological disease, and cardiovascular disorders). Definition of fall: not reported. | Community-dwelling elderly: 38 fallers (≥1 fall in past year; 10 males; 74.8 ± 5.7 years). 38 non-fallers (74.5 ± 5.0 years). | Subjects walked at preferred gait speed along a pressure-sensing walkway. Three trials were recorded for each subject. SD was calculated. | Gait speed (m/s): 1.01 vs. 1.09; swing phase (%): 36.6 vs. 37.7; stance phase (%): 63.4 vs. 62.4; double support (%): 26.6 vs. 24.5; step time variability (SD; s): 0.04 vs. 0.02; step length (m): 0.54 vs. 0.57; time peak force at maximal weight acceptance and mid-stance (s): 0.22 vs. 0.19; 0.37 vs. 0.34. | Single support (%): 36.9 vs. 37.5; step time (s): 0.55 vs. 0.53; step length variability (SD; cm): 2.6 vs. 2.7; foot progression angle (°): 8.0 vs. 6.5; peak force at maximal weight acceptance, mid-stance, and push-off (N/kg): 1.07 vs. 1.07; 0.82 vs. 0.81; 1.02 vs. 1.01; time to reach push-off (s): 0.56 vs. 0.55. |
| Marques et al., 2018 [85] | Inclusion criteria: 60–80 years; no use of any gait assistive device; no history of progressive or non-progressive neurological disease; normal cognition; normal or corrected vision; and no cardiovascular, metabolic, or musculoskeletal dysfunction that could impact the safe performance of data collection. Definition of fall: not reported. | Community-dwelling elderly: 53 fallers (≥1 fall in past year; 67.6 ± 5.3 years). 49 non-fallers (64.5 ± 7.1 years). | Subjects walked at preferred gait speed along a 14 m walkway. Six to ten trials were recorded until the subjects completed 50 consecutive strides. Two footswitches were attached on the heel and on the base of the first metatarsus (1000 Hz). | Gait speed (m/s): 1.01 vs. 1.12; stance time (s): 0.58 vs. 0.48; swing time (s): 0.48 vs. 0.57; stride time (s): 1.11 vs. 1.02; double-support time (s): 0.15 vs. 0.10; stride length (m): 1.02 vs. 1.16; stance time variability (SD; s): 0.12 vs. 0.05. | Swing time variability (SD; s): 0.25 vs. 0.17; stride time variability (SD; s): 0.23 vs. 0.21. |
| Thompson et al., 2018 [88] Qiao et al., 2018 [87] | Exclusion criteria: body mass index ≥ 30 kg/m$^2$; sedentary lifestyle; orthopedic or neurological condition; or taking medication that causes dizziness. Definition of fall: unintentionally coming to the ground or some lower level for reasons other than a violent blow, loss of consciousness, or sudden onset of paralysis. | Community-dwelling elderly: 11 fallers (≥1 fall in past year; 4 males; 78.3 ± 7.6 years). 11 non-fallers (5 males; 75.3 ± 5.4 years). | Subjects walked at preferred gait speed on a treadmill (in an immersive virtual environment). Kinematic data were assessed using a 14-camera optoelectronic motion analysis system (100 Hz). EMG of leg was recorded (1000 Hz). | | Step length (m): 0.62 vs. 0.64; step length variability (SD; cm): 2.7 vs. 2.7; step width (m): 11.2 vs. 12.9; step width variability (SD; cm): 3.3 vs. 2.8. Lower leg antagonist co-activation. |
| Watanabe, 2018 [89] | Exclusion criteria: women. Inclusion criteria: no history of any musculoskeletal or neurological disorder. Definition of fall: not reported. | Community-dwelling elderly males: 6 fallers (1 fall in past year; 69.0 ± 3.7 years). 7 non-fallers (73.3 ± 6.5 years). | Subjects walked on a treadmill at preferred gait speed for 20 min. Lower extremity kinematics were collected using a 6-camera optoelectronic motion analysis system (100 Hz). EMG data (rectus femoris muscle) were recorded. MTC was analyzed. | MTC decreased with time in non-fallers but not in fallers. | Preferred gait speed (m/s): 1.28 vs. 1.33; 5–10 min: cadence (steps/min): 118 vs. 120; toe off (% gait cycle): 63.2 vs. 61.1; MTC (% gait cycle): 84.3 vs. 82.4. 15–20 min: cadence (steps/min): 116 vs. 117; toe off (% gait cycle): 63.3 vs. 61.2; MTC (% gait cycle): 84.0 vs. 82.3. Variability in the rectus femoris activation decreased with time in fallers and non-fallers. |

**Table 1.** *Cont.*

| Study | Inclusion and/or Exclusion Criteria Definition of Fall | Sample Characteristics | Gait Assessment | Gait Parameters Related to Falls (Fallers vs. Non-Fallers) | Gait Parameters Not Related to Falls (Fallers vs. Non-Fallers) |
|---|---|---|---|---|---|
| Bueno et al., 2019 [90] | Inclusion criteria: woman; ≥65 years; independent walking without aids; absence of previous surgeries in lower limbs, pelvis, or spine; Mini Mental Status Examination score > 14; body mass index ≥ 30 kg/m$^2$; rheumatoid arthritis; neuromuscular or neurodegenerative diseases; diabetes mellitus; and no visual impairment. Definition of fall: unexpected event in which the subject comes to rest on the ground or lower level (excluded coming to rest against furniture or wall). | Community-dwelling elderly women: 10 recurrent fallers (≥2 falls in past year; 71.0 ± 6.8 years). 12 fallers (1 fall in past year; 72.7 ± 5.6 years). 27 non-fallers (72.5 ± 6.8 years). | Subjects walked barefoot at preferred gait speed along a 9 m walkway. Kinematic data were collected using a 7-camera optoelectronic motion analysis system (120 Hz). | | Fallers vs. Recurrent fallers vs. Non-fallers: gait speed (m/s): 1.02 vs. 0.99 vs. 1.00; cadence (step/min): 110 vs. 113 vs. 109; stride length (m): 1.10 vs. 1.04 vs. 1.02; hip flexion/extension (°): 8.2 vs. 7.1 vs. 8.2; knee flexion/extension (°): 7.1 vs. 8.1 vs. 7.6; ankle dorsiflexion/plantarflexion (°): 4.7 vs. 4.5 vs. 4.5; hip adduction/abduction (°): 4.7 vs. 5.2 vs. 4.2; hip rotation (°): 6.4 vs. 6.9 vs. 6.6; foot progression angle (°): 5.8 vs. 6.1 vs. 6.0. |
| Mak et al., 2019 [92] | Exclusion criteria: Mini Mental Status Examination score < 24; neurological impairment; acquired static visual acuity worse than 20/40; or unable to walk independently indoors. Definition of fall: not reported. | Community-dwelling elderly (40 males; 70.3 ± 4.8 years). 37 fallers (≥1 fall in past year; 7 males; 70.7 ± 5.0 years). 97 non-fallers (33 males; 70.1 ± 4.7 years). | Subjects walked along a 6 m walkway at preferred gait speed. Average muscle activity indices of lower limb co-contractions were measured using surface EMG. | Shank and thigh muscle co-contractions were higher in fallers. Lower limb muscle co-contractions were higher in fallers. | |
| Gillain et al., 2019 [91] | Exclusion criteria: ≥1 fall in the past year; use of a walking aid; gait disorders or an increased fall risk related to neurological or osteoarticular diseases; dementia; hip or knee prosthesis in the previous year; pain when walking; acute respiratory or cardiac illness; recent hospitalization; untreated or uncontrolled comorbidities; use of neuroleptic and sedative drugs; or presence of a cardiac pacing device. Definition of fall: unexpected event in which the subject comes to rest on the ground. | Community-dwelling elderly (48 males; 71.3 ± 5.4 years): 35 fallers (≥1 fall in 2 years follow-up; 18 males; 72.0 ± 6.9 years). 61 non-fallers (30 males; 70.9 ± 4.3 years). | Subjects walked with their own shoes at preferred and fast gait speed. Kinematic data were collected using a tri-axial accelerometer attached to the lumbar position and a 24-camera optoelectronic motion analysis system (120 Hz). Variability was analyzed using the CV. MTC was analyzed. | Preferred gait speed: stride length (m): 1.30 vs. 1.37. Fast gait speed: gait speed (m/s): 1.64 vs. 1.74; stride length (m): 1.47 vs. 1.60; | Preferred gait speed: gait speed (m/s): 1.24 vs. 1.31. Preferred gait speed and fast gait speed: cadence (stride/s): 0.96 vs. 0.96; 1.10 vs. 1.08; MTC (mm): 17.3 vs. 17.8; 18.0 vs. 20.6; median MTC (mm): 17.4 vs. 17.7; 18.7 vs. 20.8; MTC variability (SD; mm): 5.0 vs. 4.1; 4.5 vs. 4.5; MTC variability (CV; %): 27.0 vs. 24.3; 26.1 vs. 27.6; MTC-minimum (mm): 10.8 vs. 12.0; 9.8 vs. 9.1. |

**Table 1.** *Cont.*

| Study | Inclusion and/or Exclusion Criteria Definition of Fall | Sample Characteristics | Gait Assessment | Gait Parameters Related to Falls (Fallers vs. Non-Fallers) | Gait Parameters Not Related to Falls (Fallers vs. Non-Fallers) |
|---|---|---|---|---|---|
| Yamagata et al., 2019 [93] | Exclusion criteria: neurological disorders or musculoskeletal injuries that would affect performance or inability to walk without assistance. Definition of fall: an unexpected event during which the subjects come to rest on the ground or other lower level. | Community-dwelling elderly: 12 fallers ($\geq$1 fall in 1 year follow-up; 78.0 $\pm$ 4.7 years). 16 non-fallers (73.8 $\pm$ 7.9 years). | Subjects walked at preferred gait speed on a 6 m walkway. Kinematic data were assessed by an 8-camera optoelectronic motion analysis system (100 Hz). CoM was calculated. SD analyzed variability. | Gait speed (m/s): 1.1 vs. 1.3; CoM distance at toe-off and at heel strike (cm): 22.9 vs. 25.3; 19.7 vs. 2.26; variability in right shank angle in the sagittal plane during mid-swing (SD; rad): 6.41 vs. 5.21. | Segment angles (foot, shank, thigh, pelvis); segment angles variability (foot, shank, thigh, pelvis). Step width during early swing, mid-swing, and late-swing (m): 0.13 vs. 0.14; 0.15 vs. 0.17; 0.14 vs. 0.16; step width variability during early swing, mid-swing, and late-swing (m): 0.018 vs. 0.023; 0.011 vs. 0.0127; 0.019 vs. 0.020. |
| Yamagata et al., 2019 [94] | Exclusion criteria: neurological disorders or musculoskeletal injuries that would affect performance or inability to walk without assistance. Definition of fall: an unexpected event during which the subjects come to rest on the ground or other lower level. | Community-dwelling elderly: 10 fallers ($\geq$1 fall in past year; 78.0 $\pm$ 2.7 years). 14 non-fallers (75.1 $\pm$ 5.4 years). | Subjects walked at preferred gait speed on a 6 m walkway. Kinematic data were assessed by an 8-camera optoelectronic motion analysis system (100 Hz). SD is used to analyze variability. Uncontrolled manifold analysis was used to assess how variability in segmental configurations affects the frontal trajectory of the swing foot. | Step length (m): 0.52 vs. 0.56; right shank angle in early swing (lower fallers); shank angle variability during early and mid-swing (higher fallers); variability of the vertical distance between feet (higher fallers). Fallers: higher variability in segmental configurations in all phases; in ML direction, kinematic synergy was higher during the early and late swing; higher kinematic synergy in the vertical direction. | Gait speed (m/s) (statistical tendency for the difference, $p = 0.060$): 1.01 vs. 1.25; step width (m): 0.11 vs. 0.10; cadence (steps/min): 97 vs. 99; swing phase (%): 36.7 vs. 35.2; stride length ratio: 0.9 vs. 0.9. In all planes, there were no differences in segment angles (foot, shank, thigh, pelvis); segment angle variability (foot, shank, thigh, pelvis). Vertical and ML distances between feet. Variability ML distance between feet. |
| Gonzalez et al., 2020 [95] | Inclusion criteria: ability to walk one mile at any pace with minimum rest and body mass index = 18.5–24.9 kg/m$^2$ or $\geq$30 kg/m$^2$. Exclusion criteria: use of an assistive device for walking; artificial joint replacement; history of diabetic peripheral neuropathy, neurological conditions that interfere with gait; body mass index = 25–29.9 kg/m$^2$; compromised range of motion in the lower limb or trunk; untreated hypertension or cardiovascular diseases; or T-score $\leq$ 2.5 for the femoral neck. Definition of fall: not reported. | Community-dwelling elderly: 16 fallers ($\geq$1 fall in 1-year follow-up). 16 non-fallers. | Subjects walked on a treadmill for 10 min at preferred gait speed. Kinematic data of the 10th thoracic vertebra were assessed by an 8-camera optoelectronic motion analysis system (100 Hz). Short-term exponents were used to analyze the data variability for each direction (larger positive exponents indicate higher instability). | | Short-term exponents of the ML, AP, or vertical displacements. |

**Table 1.** *Cont.*

| Study | Inclusion and/or Exclusion Criteria Definition of Fall | Sample Characteristics | Gait Assessment | Gait Parameters Related to Falls (Fallers vs. Non-Fallers) | Gait Parameters Not Related to Falls (Fallers vs. Non-Fallers) |
|---|---|---|---|---|---|
| Pol et al., 2021 [97] | Exclusion criteria: <60 years; Short Portable Mental Status Questionnaire score < 7; unable to ambulate for at least 10 m without an assistive device; diabetic foot syndrome; neurological diseases; or lower extremity surgery. Definition of fall: unintentionally coming to the ground or other lower surface, not as a result of a major intrinsic event or an overwhelming hazard. | Community-dwelling elderly: 74 fallers (≥1 fall in past year; 20 males; 71.9 ± 4.9 years). 113 non-fallers (61 males; 69.9 ± 5.5 years). | Three trials were recorded for each subject's dominant limb using a two-step gait initiation protocol at a comfortable walking speed. Foot function was assessed using barefoot plantar pressure analysis (50 Hz). CoP was calculated. | CoP excursion index (%): 14.69 vs. 17.58, total pressure–time integral (% body weight * second/cm$^2$): 3.75 vs. 3.23; pressure–time integral of medial forefoot (% body weight * second/cm$^2$): 1.84 vs. 1.39. | Pressure–time integral of medial heel (% body weight * second/cm$^2$): 1.52 vs. 1.44; pressure–time integral of lateral heel (% body weight * second/cm$^2$): 1.48 vs. 1.36; pressure–time integral of medial midfoot (% body weight * second/cm$^2$): 0.99 vs. 0.87; pressure–time integral of lateral midfoot (% body weight * second/cm$^2$): 1.30 vs. 1.21; pressure–time integral of central forefoot (% body weight * second/cm$^2$): 1.73 vs. 1.71; pressure–time integral of lateral forefoot (% body weight * second/cm$^2$): 1.75 vs. 1.79; pressure–time integral of hallux (% body weight * second/cm$^2$): 2.35 vs. 1.80; pressure–time integral of second toe (% body weight * second/cm$^2$): 0.90 vs. 0.78; pressure–time integral of lateral toes (% body weight * second/cm$^2$): 0.84 vs. 0.73; Regional CoP velocity–heel, midfoot, forefoot, and toes (cm/s): 25.6 vs. 28.1; 16.1 vs. 18.8; 19.1 vs. 18.1; 43.8 vs. 48.4. |
| Sadeghi et al., 2021 [98] | Exclusion criteria: need help for walking; difficulties in understanding instructions; or receiving hospice care. Definition of fall: not reported. | Community-dwelling elderly: 13 fallers (>1 fall in past year; 72.5 ± 7.1 years). 13 non-fallers (73.1 ± 7.1 years). | Subjects walk barefoot at preferred gait speed on a 10 m walkway. Kinematic data from 10 gait cycles were collected by a 10-camera optoelectronic motion analysis system (100 Hz). | Cadence (steps/min): 98 vs. 115; gait speed (m/s): 0.74 vs. 1.04; stride time (s): 1.27 vs. 1.05; stride length (m): 0.90 vs. 1.08; double support time (s): 0.30 vs. 0.26. Ankle-to-knee, knee-to-hip, and ankle-to-hip coordination patterns. Less coordination variability in fallers. | Step width (m): 0.10 vs. 0.10; ankle displacement (°): 25 vs. 23; knee displacement (°): 47 vs. 44; hip displacement (°): 42 vs. 41. |
| Yamagata et al., 2021 [96] | Exclusion criteria: neurological disorders or musculoskeletal injuries that would affect performance or inability to walk without assistance. Definition of fall: an unexpected event during which the subjects come to rest on the ground or other lower level. | Community-dwelling elderly: 10 fallers (≥1 fall in past year; 78.0 ± 2.7 years). 14 non-fallers (75.1 ± 5.4 years). | Subjects walked at preferred gait speed on a 6 m walkway. Kinematic data were assessed by an 8-camera optoelectronic motion analysis system (100 Hz). CoM was calculated. Variance is used to analyze variability. | CoM vertical direction variability (higher fallers). | CoM displacements. CoM ML direction variability. |

**Table 1.** *Cont.*

| Study | Inclusion and/or Exclusion Criteria Definition of Fall | Sample Characteristics | Gait Assessment | Gait Parameters Related to Falls (Fallers vs. Non-Fallers) | Gait Parameters Not Related to Falls (Fallers vs. Non-Fallers) |
|---|---|---|---|---|---|
| Figueiredo et al., 2022 [99] | Inclusion criteria: ≥80 years; any gender; ability to walk independently; and ability to understand the verbal commands to carry out assessment. Exclusion criteria: uncertain about their history of falls; had been hospitalized for >7 days in the 3 months before assessment; or had a severe orthopedic, neurological, respiratory, cardiovascular, visual, or hearing disease. Definition of fall: unexpected and unexplained event in which the subject inadvertently comes to the ground. | Community-dwelling elderly: 32 fallers (≥1 fall in past 6 months; 7 males; 89.9 ± 4.4 years). 32 non-fallers (5 males; 88.6 ± 4.1 years). | Subjects walked during the Timed Up and Go test. Kinematic data were collected using a tri-axial accelerometer attached between the L5 and S1 vertebrae. Spectral arc length metrics are used to quantify gait smoothness. | | Spectral arc length in AP, ML, and vertical directions. |
| Nascimento et al., 2022 [100] | Inclusion criteria: residing in the community; 60–79 years; and able to walk independently. Exclusion criteria: medical contraindications for submaximal exercise, according to American College of Sports Medicine guidelines, or inability to understand or follow investigation protocols. Definition of fall: not reported. | Community-dwelling elderly: 225 fallers (>1 fall in past year; 70.1 ± 5.6 years). 394 non-fallers (69.2 ± 5.4 years). | Subjects walked a distance of 30 feet at their preferred gait speed. Gait speed is calculated by dividing the distance walked by the time needed. Cadence is calculated by dividing the number of steps taken in space during the 30-foot walk test by the time taken to cover that distance. | Gait speed (m/s): 1.20 vs. 1.28. | Cadence (steps/s): 1.90 vs. 1.92. |
| Yoshida et al., 2022 [101] | Exclusion criteria: diagnosis of dementia, recent major illness, neurological, sensory, or mobility impairment, or musculoskeletal disorders or injuries. Definition of fall: event that resulted in a person coming to rest unintentionally on the ground or other lower level, not as the result of a major intrinsic event or an overwhelming hazard. | Community-dwelling elderly: 28 fallers (≥1 fall in past year; 7 males; 77.5 ± 4.9 years). 28 non-fallers (12 males; 78.1 ± 5.3 years). | Gait initiation was assessed using two force platforms (1024 Hz). CoP was calculated. | First step length (m): 0.61 vs. 0.66; CoP ML displacement during weight transfer (m): 0.13 vs. 0.12. | Weight transfer time (s): 0.99 vs. 0.96; forward progression time (s): 0.48 vs. 0.49; first step time (s): 0.55 vs. 0.57; ground contact time (s): 0.69 vs. 0.68; first step width (m): 0.17 vs. 0.17; CoP AP displacement during weight transfer (m): 0.043 vs. 0.048; CoP AP displacement during forward progress (m): 0.181 vs. 0.170; CoP ML displacement during forward progress (m): 0.023 vs. 0.023; CoP AP displacement during first step (m): 0.228 vs. 0.225; CoP ML displacement during first step (m): 0.023 vs. 0.021. |

**Table 1.** *Cont.*

| Study | Inclusion and/or Exclusion Criteria Definition of Fall | Sample Characteristics | Gait Assessment | Gait Parameters Related to Falls (Fallers vs. Non-Fallers) | Gait Parameters Not Related to Falls (Fallers vs. Non-Fallers) |
|---|---|---|---|---|---|
| Baba et al., 2023 [102] | Inclusion criteria: ≥65 years and ability to walk independently without using aids. Exclusion criteria: stroke diagnosis, Parkinson's disease, rheumatism; or history of hip or knee surgery. Definition of fall: unintentional landing on the ground, floor, or lower level. | Community-dwelling elderly: 16 fallers (≥1 fall in past year; 4 males; 84.6 ± 5.7 years). 34 non-fallers (3 males; 81.7 ± 6.1 years). | Subjects walked barefoot on a walkway at preferred gait speed. Kinematic data were collected using an inertial sensor system. | Gait speed (m/s): 0.83 vs. 0.92; foot angle with ground (°): 13.6 vs. 18.3. | Stride time (s): 1.10 vs. 1.03; stride length (m): 0.87 vs. 0.96; cadence (steps/min): 114 vs. 117; stance phase (%): 65.0 vs. 64.7; single support (%): 35.4 vs. 35.9; double support (%): 21.2 vs. 21.0; maximum ankle plantarflexion: 10.8 vs. 9.0; maximum ankle dorsiflexion: 9.7 vs. 9.4; maximum knee flexion: 40.4 vs. 41.7; maximum knee extension: 0.5 vs. 0.5; maximum hip flexion: 25.2 vs. 27.3; maximum hip extension: 13.5 vs. 13.2. |

AP—anteroposterior; ApEn—approximate entropy analysis; CoM—center of mass; CoP—center of pressure; CV—coefficient of variation; EMG—electromyography; ML—mediolateral; MTC—minimum toe clearance; rMSSD—root-mean-square of differences between intervals; SampEn—sample entropy analysis; SD—standard deviation; SDANN—standard deviation of means of intervals taken every five strides; SDNN—standard deviation of all time intervals; SDNNi—mean of standard deviations of intervals. *—it is the symbol of multiplication.

**Table 2.** Characteristics and data of the studies that compared elderly who fell during induced falls and elderly who did not.

| Study | Inclusion and/or Exclusion Criteria Definition of Fall | Sample Characteristics | Gait Assessment | Gait Parameters Related to Falls (Fallers vs. Recoveries) | Gait Parameters Not Related to Falls (Fallers vs. Recoveries) |
|---|---|---|---|---|---|
| Pavol et al., 1999 [105] | Exclusion criteria: neurological, musculoskeletal, cardiovascular, pulmonary, cognitive, and other systemic disorders; history of repeated falling; or minimum bone mineral density of the femoral neck of 0.65 g/cm². | Community-dwelling elderly: 10 fallers (body's subject being at least 50% supported by the safety harness). 39 recoveries. | Subjects walked on a 7 m walkway at their preferred gait speed using a safety harness. One trip was induced using a mechanical obstacle (5.1 cm from the ground). Kinematic data were collected using a 6-camera optoelectronic motion analysis system (60 Hz). | Gait speed (m/s): 1.31 vs. 1.18; step time (s): 0.50 vs. 0.54; step length (% body mass): 43.1 vs. 39.9. | Step width (cm): 8.9 vs. 9.2; trunk flexion (°): 11 vs. 9; gait phase of trip (% stride length): 59 vs. 59. |
| Lockhart et al., 2003 [106] | Exclusion criteria: based on an exam conducted by a physician to ensure that they were in generally good physical health and a peripheral neuropathy examination (below 50% of the norm). | Community-dwelling elderly (7 males: 75.5 ± 6.8 years): 7 fallers. 7 non-fallers. | Subjects walked on a 20 m circular track for 10 min using a safety harness. Two slips were induced. Four video cameras and two force plates were used to collect 3D data and the ground reaction forces (60 Hz). | Coefficient of friction after a heel strike (horizontal ground reaction force/vertical ground reaction force) was lower in fallers; horizontal heel contact velocity was higher in fallers. | |

**Table 2.** *Cont.*

| Study | Inclusion and/or Exclusion Criteria Definition of Fall | Sample Characteristics | Gait Assessment | Gait Parameters Related to Falls (Fallers vs. Recoveries) | Gait Parameters Not Related to Falls (Fallers vs. Recoveries) |
|---|---|---|---|---|---|
| Pijnappels et al., 2005 [103] | Exclusion and inclusion criteria: not reported. | Community-dwelling elderly: 7 fallers (when the vertical force in the ropes exceeded 200 N during trials when one obstacle appeared from the ground unexpectedly to catch the subject's swing limb. 1 male; 67.9 ± 2.6 years). 4 non-fallers (3 males; 66.5 ± 3.3 years). | Subjects walked at preferred gait speed over a platform and were tripped several times by an obstacle that appeared from the floor. A safety harness prevented subjects from falling. Kinematic and ground reaction force data were collected using a 4-camera optoelectronic motion analysis system and a force plate (100 Hz). | Total angular momentum; angular momentum at push-off. | Gait speed; stride length; obstacle contact phase; stance time of the support limb; double support time; swing phase time of the recovery limb; hip horizontal displacement; hip height at end push-off; rate of change of moment generation in ankle, knee, and hip; hip extension moment, knee flexion moment and ankle plantarflexion moment peak in the support limb. |
| Pavol et al., 2001 [107] | Exclusion criteria: neurological, musculoskeletal, cardiovascular, pulmonary, cognitive, and other systemic disorders; history of repeated falling; or minimum bone mineral density of the femoral neck of 0.65 g/cm². | Community-dwelling elderly. The recovery attempts were classified as a lowering or an elevating strategy. Elevating strategy: 1 faller. 11 recoveries. | Subjects walked on a 7 m walkway at their preferred gait speed using a safety harness. One trip was induced using a mechanical obstacle (5.1 cm from the ground). Kinematic collected using a 6-camera optoelectronic motion analysis system (60 Hz). Ground reaction forces were collected by two force plates (1000 Hz). | Hip horizontal velocity at time of trip (% body height/s): 92.5 vs. 68.0; hip horizontal velocity at 0.1 s post-trip (%/s): 86.5 vs. 67.4; time from trip to follow-through toe-off (s): 0.40 vs. 0.45; ankle–hip angular position at time of loading (°): 11.9 vs. 8.9; hip height at heel strike (% body height): 51.1 vs. 54.5; trunk inclination from vertical at heel strike (°): 58.5 vs. 37.3; lumbar flexion at heel strike (°): 45.2 vs. 23.1; minimum hip–ankle distance (% body height): 42.4 vs. 47.4; maximum trunk inclination from vertical (°): 83.5 vs. 49.7; maximum lumbar flexion at heel strike (°): 70.3 vs. 35.3. | Trunk inclination at time of trip (°): 14.3 vs. 8.7; hip vertical velocity 0.1 s post-trip (% body height): −8.1 vs. −9.3; lumbar flexion at time of loading (°): 17.2 vs. 6.7; recovery step length (% body height): 51.8 vs. 49.8; recovery stride length (% body height): 93.2 vs. 89.7; obstacle–ankle distance at heel strike (% body height): 32.6 vs. 32.2; minimum hip–ankle distance (% body height): 31.0 vs. 34.5; maximum ankle ground clearance (% body height): 24.0 vs. 22.1; time from trip-to-ground contact (s): 0.40 vs. 0.45; maximum horizontal ankle velocity (% body height/s): 203 vs. 225; average horizontal ankle velocity (% body height/s): 56 vs. 54; ankle–hip angle at heel strike (°): 0.3 vs. −7.6; maximum hip vertical velocity (% body height/s): 20.7 vs. 29.1. |
| Pavol et al., 2001 [107] | Exclusion criteria: neurological, musculoskeletal, cardiovascular, pulmonary, cognitive, and other systemic disorders; history of repeated falling; or minimum bone mineral density of the femoral neck of 0.65 g/cm². | Community-dwelling elderly. The recovery attempts were classified as a lowering or an elevating strategy. Lowering strategy: 5 fallers during-step fall (body's subject being at least 50% supported by safety harness). 3 fallers after-step fall (body's subject being at least 50% supported by safety harness). 26 recoveries. | Subjects walked on a 7 m walkway at their preferred gait speed using a safety harness. One trip was induced using a mechanical obstacle (5.1 cm from the ground). Kinematic data were collected using a 6-camera optoelectronic motion analysis system (60 Hz). Ground reaction forces were collected by two force plates (1000 Hz). | Fallers during-step fall vs. recoveries: hip horizontal velocity at trip (% body height/s): 91.3 vs. 68.2; hip horizontal velocity at 0.1 s post-trip (% body height/s): 94.5 vs. 72.9; time from trip to support limb loading (s): 0.27 vs. 0.16; ankle–hip angle at time of loading (°): 23.6 vs. 9.8; recovery step length (% body height): 36.9 vs. 49.4; | Fallers during-step fall vs. recoveries: trunk inclination at time of trip (°): 7.5 vs. 9.1; hip vertical velocity at 0.1 s post-trip (body height/s): −11.8 vs. −9.8; time from trip to follow-through toe-off (s): 0.49 vs. 0.50; lumbar flexion at time of loading (°): 6.4 vs. 6.1; minimum hip–ankle distance at recovery step (% body height): 31.8 vs. 33.0; maximum ankle ground clearance (% body height): 23.8 vs. 24.7; time from trip to recovery |

**Table 2.** *Cont.*

| Study | Inclusion and/or Exclusion Criteria Definition of Fall | Sample Characteristics | Gait Assessment | Gait Parameters Related to Falls (Fallers vs. Recoveries) | Gait Parameters Not Related to Falls (Fallers vs. Recoveries) |
|---|---|---|---|---|---|
| | | | | recovery stride length (% body height): 51.4 vs. 59.9; obstacle–ankle distance at ground contact (% body height): 32.0 vs. 39.6; recovery step time (s): 0.21 vs. 0.27; hip height at ground foot contact (% body height): 47.2 vs. 54.5; ankle–hip angle at heel strike (°): 12.7 vs. −10.1; trunk inclination from vertical at heel strike (°): 48.3 vs. 36.0; maximum lumbar flexion at heel strike (°): 23.1 vs. 35.6. Fallers after-step fall vs. recoveries: trunk inclination at trip (°): 18.8 vs. 9.1; trunk inclination from vertical at ground foot contact (°): 55.2 vs. 36.0; lumbar flexion at ground foot contact (°): 38.7 vs. 23.5; maximum hip vertical velocity (% body height): −0.4 vs. 32.2; minimum hip–ankle distance (% body height): 41.0 vs. 47.3; maximum trunk inclination from vertical at ground foot contact (°): 74.6 vs. 46.6; maximum lumbar flexion at ground foot contact (°): 54.4 vs. 35.6. | foot toe-off (s): 0.28 vs. 0.26; time from trip to ground contact (s): 0.49 vs. 0.52; maximum horizontal ankle velocity (% body height/s): 227 vs. 263; average horizontal ankle velocity (% body height/s): 109 vs. 115; lumbar flexion at ground foot contact (°): 22.1 vs. 23.5; Fallers after-step fall vs. recoveries: hip horizontal velocity at time of trip (% body height/s): 79.4 vs. 68.2; hip horizontal velocity at 0.1 s post-trip (% body height/s): 82.2 vs. 72.9; hip vertical velocity at 0.1 s post-trip (% body height/s): −7.2 vs. −9.8; time from trip to support limb loading (s): 0.14 vs. 0.16; time from trip to follow-through toe-off (s): 0.52 vs. 0.50; ankle–hip angle at time of loading (°): 9.1 vs. 9.8; lumbar flexion at time of loading (°): 15.4 vs. 6.1; recovery step length (% body height): 49.1 vs. 49.4; recovery stride length (% body height): 61.7 vs. 59.9; obstacle–ankle distance at heel strike (% body height): 40.0 vs. 39.6; minimum hip–ankle distance (% body height): 28.8 vs. 33.0; time from trip to recovery foot toe-off (s): 0.24 vs. 0.26; time from trip to heel strike (s): 0.51 vs. 0.52; recovery step duration (s): 0.26 vs. 0.270; maximum horizontal ankle velocity (% body height/s): 264 vs. 263; average horizontal ankle velocity (% body height/s): 117 vs. 115; hip height (% body height): 50.9 vs. 54.5, ankle–hip angle (°): −7.8 vs. −10.1. |
| Espy & Pai, 2007 [108] | Inclusion criteria: ≥65 years. | Community-dwelling elderly: 19 fallers (subjects who fell during an unexpected induced slip). 15 non-fallers (subjects who recover during an unexpected induced slip). | Subjects walked on an instrumented walkway while wearing a safety harness. After 10 unperturbed trials, an unexpected slip was induced under the right heel. Motion data were used to calculate frontal plane variables. | | Step width; CoM lateral position; CoM velocity. |

**Table 2.** *Cont.*

| Study | Inclusion and/or Exclusion Criteria Definition of Fall | Sample Characteristics | Gait Assessment | Gait Parameters Related to Falls (Fallers vs. Recoveries) | Gait Parameters Not Related to Falls (Fallers vs. Recoveries) |
|---|---|---|---|---|---|
| Bhatt et al., 2011 [104] | Exclusion criteria: Folstein Mini Mental Status Examination score < 25 or classified as osteopenic or osteoporotic. | Community-dwelling elderly (44 males): 59 fallers (the force recorded on the safety harness load cell force exceeded 30% of the body weight; 71.6 ± 4.6 years). 56 recoveries (71.4 ± 5.1 years). | Subjects walked for 9–12 trials on a 7 m walkway using their own athletic shoes and were then exposed to one unannounced slip. Kinematic and ground reaction force data were collected during 5 trials using an 8-camera optoelectronic motion analysis system (120 Hz) and one force platform (600 Hz). | Dynamic gait stability: −0.16 vs. −0.13. | |
| Yang & Pai, 2014 [109] | Exclusion criteria: any known neurological, musculoskeletal, or other systemic disorder that would have affected their postural control. | Community-dwelling elderly: 98 fallers (the force recorded on the safety harness load cell force exceeded 30% of the body weight; 22 males; 71.8 ± 5.5 years). 89 recoveries (37 males; 71.9 ± 4.8 years). | Subjects walked for 20 trials on a 7 m instrumented walkway at preferred gait speed and were then exposed to one unannounced slip. Kinematic data were collected using an 8-camera optoelectronic motion analysis system (120 Hz) and ground reaction force using four force platforms (600 Hz). | Step width (SD; m): 0.031 vs. 0.027; dynamic stability of CoM against backward falling: −0.18 vs. −0.16. | Step length (SD; m): 0.070 vs. 0.062; step time (SD; s): 0.044 vs. 0.041; margin of stability: 0.039 vs. 0.051; Floquet multiplier: 0.422 vs. 0.432; Lyapunov exponent (short-term): 0.671 vs. 0.737; Lyapunov exponent (long-term): 0.034 vs. 0.026. |
| Sawers et al., 2016 [110] | Inclusion criteria: participants who experienced a "split" slip with the slipping and trailing feet traveling apart were included. | Community-dwelling elderly: 15 fallers (unable to regain their stability after an unexpected induced slip; 2 males; 71.0 ± 2.0 years). 13 recoveries (able to recover their stability and continue walking after an unexpected induced slip; 8 males; 72.0 ± 5.0 years). | Subjects walked on a 7 m walkway at their preferred gait speed using a safety harness. One unexpected slip was induced. Kinematic collected using an optoelectronic motion analysis system (120 Hz). EMG of TA, MG, VL, and BF were recorded (600 Hz). | Slip distance (m): 0.78 vs. 0.61; EMG onset latencies/slip leg (s): VL (right): 0.239 vs. 0.186; BF (right): 0.170 vs. 0.120. Muscle synergies recruited during slip and non-slip trials: 3.7 vs. 4.7. | Slip time (s): 0.82 vs. 0.94; peak slip velocity (m/s): 2.00 vs. 1.84; dynamic stability: −0.18 vs. −0.16; gait speed (m/s): 0.89 vs. 1.00; shank angle (°): 74 vs. 73. EMG onset latencies/slip leg (s): TA (right): 0.173 vs. 0.151; MG (right): 0.234 vs. 0.232; EMG onset latencies/nonslip leg (s): TA (left): 0.215 vs. 0.198, VL (left): 0.165 vs. 0.154; BF (left): 0.150 vs. 0.155; EMG onset peak magnitude/slip leg: TA (right): 2.30 vs. 2.28; MG (right): 2.32 vs. 2.32; VL (right): 2.09 vs. 2.36; BF (right): 3.45 vs. 3.87; EMG Onset peak magnitude/nonslip leg: TA (left): 2.86 vs. 2.58; MG (left): 1.64 vs. 1.87; VL (left): 4.75 vs. 3.56; BF (left): 3.75 vs. 3.22. |

**Table 2.** *Cont.*

| Study | Inclusion and/or Exclusion Criteria Definition of Fall | Sample Characteristics | Gait Assessment | Gait Parameters Related to Falls (Fallers vs. Recoveries) | Gait Parameters Not Related to Falls (Fallers vs. Recoveries) |
|---|---|---|---|---|---|
| Sawers & Bhatt, 2018 [111] | Inclusion criteria: participants who experienced a feet-forward slip (with both feet moving forward) were included. | Community-dwelling elderly: 12 fallers (when peak force recorded by the load cell in line with the overhead harness exceeded 30% of the subject's body weight; 2 males; 73.0 ± 4.9 years). 13 recoveries (7 males; 74.0 ± 4.1 years). | Subjects walked on a 7 m walkway at their preferred gait speed using a safety harness. One unexpected slip was induced. Kinematic data were collected using an optoelectronic motion analysis system (120 Hz). Ground reaction forces were collected (600 Hz). | Lower limb joint angle: knee flexion (higher flexion fallers). Number of muscle synergies recruited: 4 vs. 5. | Peak slip velocity (m/s): 2.28 vs. 2.14; slip duration (s): 0.68 vs. 0.70; slip distance (m): 0.74 vs. 0.62; shank angle (°): 75.8 vs. 75.0; step length (m): 0.30 vs. 0.32; dynamic stability: −0.124 vs. −0.155; gait speed (m/s): 1.13 vs. 1.02 (statistical tendency for the difference, $p = 0.093$). Lower limb joint angle: hip flexion/extension; knee extension; ankle dorsiflexion/plantar-flexion. |
| Bruijn et al., 2022 [112] | Exclusion criteria: orthopedic, neuromuscular, cardiac, or visual problems. | Community-dwelling elderly: 5 fallers. 11 recoveries. | Subjects walked on a 12 m walkway at their preferred gait speed using a safety harness. One unexpected slip was induced. Kinematic data were collected using an optoelectronic motion analysis system (50 Hz). Ground reaction forces were collected (1000 Hz). | Sagittal plane forward body rotation at touchdown. | Gait speed (m/s): 1.48 vs. 1.43; time between impact and touchdown (s): 0.46 vs. 0.50; arm movements. |
| Wang et al., 2022 [113] | Inclusion criteria: ≥60 years. Exclusion criteria: recently (≤6 months) self-reported diagnosed neurological, musculoskeletal, or other systemic disorder. | Community-dwelling elderly: 229 falls (the recovery foot landing posterior to the sliding foot based on the location of heel markers). 569 recoveries. | Subjects walked with their own shoes on a 12 m walkway at their preferred gait speed using a safety harness. Unexpected slips were induced. Kinematic data were collected using an 8-camera optoelectronic motion analysis system (120 Hz). Ground reaction forces were collected (600 Hz). | Stride length (m): 0.41 vs. 0.70; slip distance (m): 0.31 vs. 0.08; slip velocity (m/s): 1.12 vs. −0.34; trunk angle (°): 4.9 vs. −1.5. | |
| Wang & Bhatt, 2023 [114] | Inclusion criteria: ≥60 years. Exclusion criteria: recently (≤6 months) self-reported diagnosed neurological, musculoskeletal, or other systemic disorder. | Community-dwelling elderly: 61 fallers (the peak moving average force of the load cell over a 1 s period was ≥4.5% of body weight). 56 recoveries. | Subjects walked with their own shoes on a 12 m walkway at their preferred gait speed using a safety harness. Unexpected slips were induced. Kinematic data were collected using an 8-camera optoelectronic motion analysis system (120 Hz). Ground reaction forces were collected (600 Hz). CoM was calculated. | Margin of stability at tripping foot touchdown: 1.39 vs. 0.95; maximum step length (m): 0.11 vs. 0.04; maximum CoM velocity (m/s): 0.59 vs. 0.21; peak trunk angle (°): 23.7 vs. 6.9. | Margin of stability at tripping foot lift-off: 1.03 vs. 0.96; margin of stability at pre-tripping foot touchdown: 2.38 vs. 2.24. |

BF—biceps femoris long head; CoM—center of mass; EMG—electromyography; MG—medial gastrocnemius; SD—standard deviation; TA—tibialis anterior; VL—vastus lateralis.

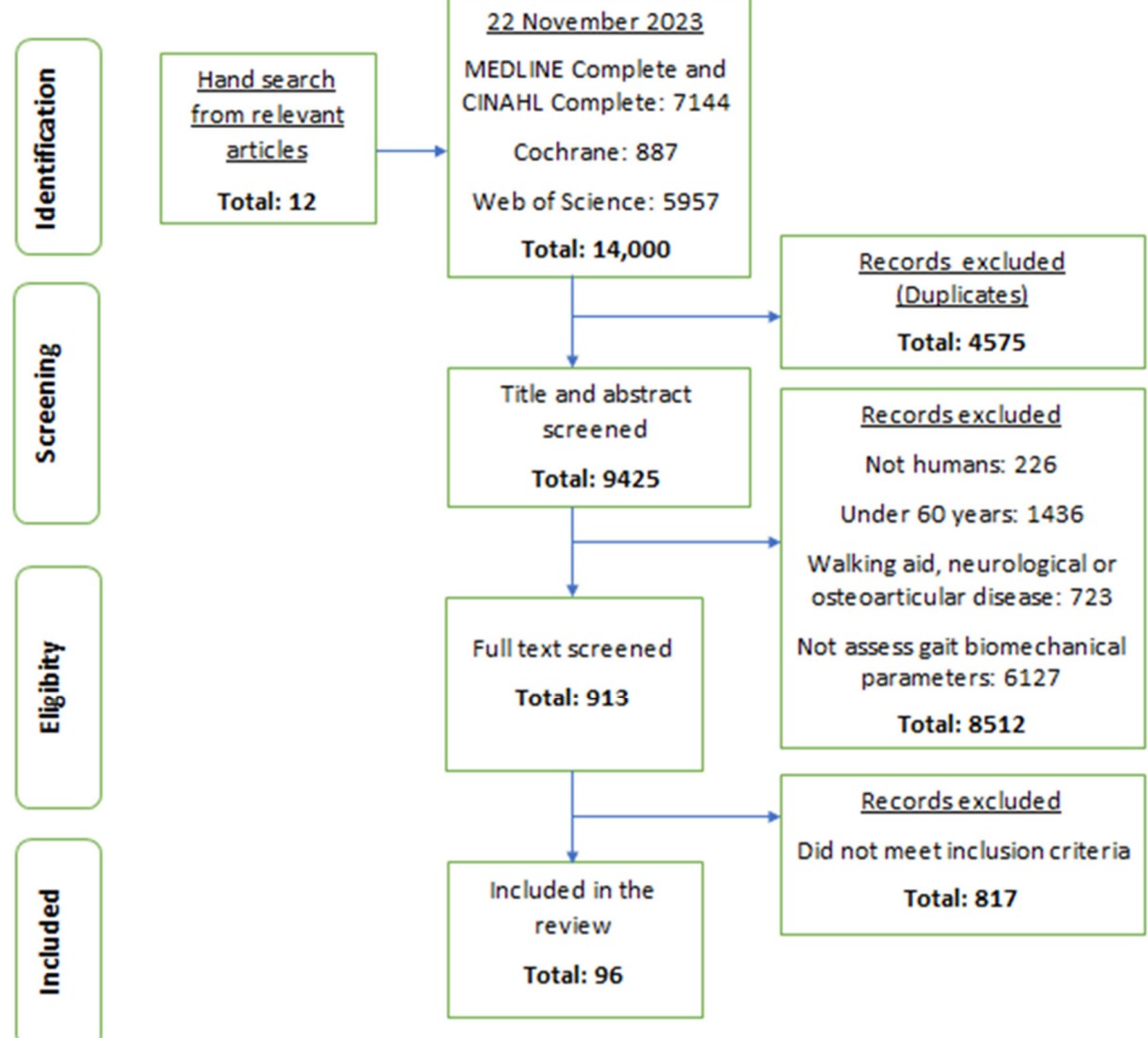

**Figure 1.** Study flow diagram.

Out of the 86 selected studies that compared elderly fallers and non-fallers, 72 (83.7%) were retrospective, and 12 (14%) were prospective; the remaining two studies compared the elderly who fell from induced falls with the elderly who recovered during unperturbed gait trials. Among the retrospective studies, fifteen evaluated the history of falls during the previous 6 months, one during the previous 10 months, forty-nine during the previous year, two during the previous 2 years, and one during the previous 5 years; four studies did not report this information. Among the prospective studies, two evaluated the occurrence of falls during a 6-month follow-up, eight during a 1-year follow-up, and two during a 2-year follow-up. Additionally, 38 of the studies (44.2%) did not provide a definition of the term "fall".

Regarding the selected retrospective studies, 50 analyzed the subjects' gait on a walk-way (forty-four during level ground, one during unleveled ground, and five during gait initiation); 19 on a treadmill (18 during level gait and 1 during gait initiation); and 3 during real-life scenarios. Concerning the prospective studies, 11 analyzed the subjects' level gait on a walkway and 1 on a treadmill.

The elderly's gait was analyzed on a walkway in all studies that induced falls during their methodological set-up (three studies induced trips and nine induced slips).

### 3.2. Risk of Bias Assessment

Out of the 96 studies, 5 studies had a global classification of moderate, and 91 studies had a global classification of weak (Table 3). Thus, 77 out of the 96 studies were classified as weak regarding the selection bias domain because the subjects were not representative of the study population, i.e., the samples were convenience samples; the remaining 19 studies were classified as moderate because the sample was representative of the population and the studies were completed by 80–100% of the initially included subjects. Regarding the study design domain, the 96 studies were classified as weak since their study design was cross-sectional and the subject selection was not randomized. Relating to the confounders domain, 78 studies were classified as weak because the potential confounders were not shown, and 18 studies were classified as strong because the potential confounders were controlled. Concerning the blinding domain, 24 studies were classified as strong because the investigators were blinded to the status of the subjects, and the subjects were also blinded to the research question, while 72 studies were classified as moderate. Regarding the data collection methods domain, two studies were classified as weak because those methods were not reliable, or the validity and reliability of the instruments were not shown; the remaining 94 studies were classified as strong because it was shown that the instruments were valid and reliable. With respect to the withdrawals and dropouts domain, 84 studies were classified as moderate because the studies were retrospective; in this component, only 12 studies were classified as strong because the percentage of subjects that completed the study was 80% or more.

**Table 3.** Methodological quality evaluation of the included studies using the Quality Assessment Tool for Quantitative Studies.

| Study | Selection Bias | Study Design | Confounders | Blinding | Data Collection Methods | Withdrawals and Dropouts | Global |
|---|---|---|---|---|---|---|---|
| Heitmann et al., 1989 [20] | 3 | 3 | 3 | 2 | 3 | 2 | 3 |
| Gehlsen & Whaley, 1990 [21] | 3 | 3 | 3 | 2 | 1 | 2 | 3 |
| Feltner et al., 1994 [22] | 3 | 3 | 3 | 2 | 1 | 2 | 3 |
| Wolfson et al., 1995 [23] | 3 | 3 | 3 | 2 | 3 | 2 | 3 |
| Maki, 1997 [24] | 3 | 3 | 1 | 1 | 1 | 1 | 3 |
| Lee & Kerrigan, 1999 [25] | 3 | 3 | 3 | 2 | 1 | 2 | 3 |
| Nelson et al., 1999 [26] | 3 | 3 | 3 | 2 | 1 | 2 | 3 |
| Pavol et al., 1999 [105] | 3 | 3 | 1 | 1 | 1 | 2 | 3 |
| Wall et al., 2000 [27] | 3 | 3 | 3 | 2 | 3 | 2 | 3 |
| Hausdorff et al., 2001 [28] | 3 | 3 | 3 | 1 | 1 | 1 | 3 |
| Kerrigan et al., 2000 [29] | 3 | 3 | 3 | 2 | 1 | 2 | 3 |
| Kerrigan et al., 2001 [30] | 3 | 3 | 3 | 2 | 1 | 2 | 3 |
| Pavol et al., 2001 [107] | 3 | 3 | 1 | 1 | 1 | 2 | 3 |
| Kemoun et al., 2002 [31] | 3 | 3 | 3 | 1 | 1 | 1 | 3 |
| Auvinet et al., 2003 [32] | 3 | 3 | 3 | 2 | 1 | 2 | 3 |
| Mbourou et al., 2003 [33] | 3 | 3 | 3 | 2 | 1 | 2 | 3 |
| Lockhart et al., 2003 [106] | 3 | 3 | 3 | 1 | 1 | 2 | 3 |
| Chiba et al., 2005 [34] | 3 | 3 | 1 | 2 | 1 | 2 | 3 |
| Pijnappels et al., 2005 [103] | 3 | 3 | 3 | 1 | 1 | 2 | 3 |
| Barak et al., 2006 [35] | 3 | 3 | 3 | 2 | 1 | 2 | 3 |
| Toulotte et al., 2006 [36] | 3 | 3 | 1 | 2 | 1 | 2 | 3 |
| Espy & Pai, 2007 [108] | 3 | 3 | 3 | 1 | 1 | 2 | 3 |
| Karmakar et al., 2007 [37] | 3 | 3 | 3 | 2 | 1 | 2 | 3 |
| Newstead et al., 2007 [39] | 3 | 3 | 1 | 2 | 1 | 2 | 3 |
| Barrett et al., 2008 [40] | 3 | 3 | 3 | 2 | 1 | 2 | 3 |
| Khandoker et al., 2008 [41] | 3 | 3 | 3 | 2 | 1 | 2 | 3 |

**Table 3.** *Cont.*

| Study | Selection Bias | Study Design | Confounders | Blinding | Data Collection Methods | Withdrawals and Dropouts | Global |
|---|---|---|---|---|---|---|---|
| Khandoker et al., 2008 [42] | 3 | 3 | 3 | 2 | 1 | 2 | 3 |
| Lockhart & Liu, 2008 [43] | 3 | 3 | 3 | 2 | 1 | 2 | 3 |
| Verghese et al., 2009 [14] | 2 | 3 | 1 | 1 | 1 | 1 | 2 |
| Greany & Di Fabio, 2010 [44] | 3 | 3 | 3 | 2 | 1 | 2 | 3 |
| Greene et al., 2010 [45] | 2 | 3 | 3 | 2 | 1 | 2 | 3 |
| Mickle et al., 2010 [46] | 2 | 3 | 1 | 1 | 1 | 1 | 3 |
| Bhatt et al., 2011 [104] | 3 | 3 | 3 | 1 | 1 | 2 | 3 |
| Kirkwood et al., 2011 [47] | 3 | 3 | 3 | 2 | 1 | 2 | 3 |
| Lázaro et al., 2011 [48] | 2 | 3 | 3 | 2 | 3 | 2 | 3 |
| Lugade et al., 2011 [49] | 3 | 3 | 3 | 2 | 1 | 2 | 3 |
| Panzer et al., 2011 [50] | 3 | 3 | 3 | 2 | 3 | 2 | 3 |
| Scanaill et al., 2011 [51] | 2 | 3 | 3 | 2 | 1 | 2 | 3 |
| Karmakar et al., 2012 [38] | 3 | 3 | 3 | 2 | 1 | 2 | 3 |
| Uemura et al., 2012 [52] | 3 | 3 | 1 | 2 | 1 | 2 | 3 |
| Chen & Chou, 2013 [53] | 3 | 3 | 3 | 2 | 1 | 2 | 3 |
| Chen et al., 2013 [54] | 3 | 3 | 3 | 2 | 1 | 2 | 3 |
| Chiu & Chou, 2013 [55] | 3 | 3 | 3 | 2 | 1 | 2 | 3 |
| Fritz et al., 2013 [56] | 3 | 3 | 3 | 2 | 1 | 2 | 3 |
| Marques et al., 2013 [58] | 3 | 1 | 3 | 2 | 1 | 2 | 3 |
| Marques et al., 2013 [59] | 3 | 1 | 3 | 2 | 1 | 2 | 3 |
| Weiss et al., 2013 [57] | 3 | 3 | 1 | 2 | 1 | 2 | 3 |
| Ayoubi et al., 2014 [60] | 2 | 3 | 3 | 2 | 1 | 2 | 3 |
| Barelle et al., 2014 [61] | 3 | 3 | 3 | 2 | 1 | 2 | 3 |
| Iwata et al., 2014 [62] | 3 | 3 | 3 | 2 | 1 | 2 | 3 |
| Kobayashi et al., 2014 [63] | 3 | 3 | 3 | 2 | 1 | 2 | 3 |
| König et al., 2014 [64] | 3 | 3 | 1 | 2 | 1 | 2 | 3 |
| Mignardot et al., 2014 [65] | 2 | 3 | 1 | 1 | 1 | 1 | 2 |
| Yang & Pai, 2014 [109] | 2 | 3 | 3 | 1 | 1 | 2 | 3 |
| Cebolla et al., 2015 [66] | 3 | 3 | 3 | 2 | 1 | 2 | 3 |
| MacAulay et al., 2015 [67] | 2 | 3 | 3 | 2 | 1 | 2 | 3 |
| Rispens et al., 2015 [68] | 3 | 3 | 3 | 2 | 1 | 2 | 3 |
| Wright et al., 2015 [69] | 3 | 3 | 3 | 2 | 1 | 2 | 3 |
| Bounyong et al., 2016 [70] | 3 | 3 | 3 | 2 | 1 | 2 | 3 |
| Fujimoto & Chou, 2016 [71] | 3 | 3 | 3 | 2 | 1 | 2 | 3 |
| Howcroft et al., 2016 [73] | 2 | 3 | 3 | 2 | 1 | 2 | 3 |
| Ihlen et al., 2016 [72] | 3 | 3 | 3 | 2 | 1 | 2 | 3 |
| Rinaldi et al., 2016 [74] | 3 | 3 | 3 | 2 | 1 | 2 | 3 |
| Sawers et al., 2016 [110] | 3 | 3 | 3 | 1 | 1 | 2 | 3 |
| Bizovska et al., 2017 [76] | 2 | 3 | 3 | 1 | 1 | 1 | 3 |
| de Melker Worms et al., 2017 [77] | 3 | 3 | 3 | 2 | 1 | 2 | 3 |
| de Melker Worms et al., 2017 [78] | 3 | 3 | 3 | 2 | 1 | 2 | 3 |
| Marques et al., 2017 [79] | 3 | 3 | 1 | 2 | 1 | 2 | 3 |
| Júnior et al., 2017 [80] | 3 | 3 | 3 | 2 | 1 | 2 | 3 |
| Rinaldi et al., 2017 [75] | 3 | 3 | 3 | 2 | 1 | 2 | 3 |
| Svoboda et al., 2017 [81] | 2 | 3 | 3 | 2 | 1 | 2 | 3 |
| Allen & Franz, 2018 [82] | 3 | 3 | 3 | 2 | 1 | 2 | 3 |
| Benson et al., 2018 [83] | 3 | 3 | 3 | 2 | 1 | 2 | 3 |
| Howcroft et al., 2018 [86] | 3 | 3 | 3 | 1 | 1 | 1 | 3 |
| Kwon et al., 2018 [84] | 3 | 3 | 3 | 2 | 1 | 2 | 3 |
| Marques et al., 2018 [85] | 3 | 3 | 1 | 2 | 1 | 2 | 3 |
| Qiao et al., 2018 [88] | 3 | 3 | 3 | 2 | 1 | 2 | 3 |
| Sawers & Bhatt, 2018 [111] | 3 | 3 | 3 | 1 | 1 | 2 | 3 |

**Table 3.** *Cont.*

| Study | Selection Bias | Study Design | Confounders | Blinding | Data Collection Methods | Withdrawals and Dropouts | Global |
|---|---|---|---|---|---|---|---|
| Thompson et al., 2018 [87] | 3 | 3 | 3 | 2 | 1 | 2 | 3 |
| Watanabe et al., 2018 [89] | 3 | 3 | 3 | 2 | 1 | 2 | 3 |
| Bueno et al., 2019 [90] | 3 | 3 | 3 | 2 | 1 | 2 | 3 |
| Gillian et al., 2019 [91] | 2 | 3 | 1 | 1 | 1 | 1 | 2 |
| Mak et al., 2019 [92] | 2 | 3 | 3 | 2 | 1 | 2 | 3 |
| Yamagata et al., 2019 [93] | 3 | 3 | 3 | 1 | 1 | 2 | 3 |
| Yamagata et al., 2019 [94] | 3 | 3 | 3 | 1 | 1 | 1 | 3 |
| Gonzalez et al., 2020 [95] | 3 | 3 | 3 | 2 | 1 | 1 | 3 |
| Pol et al., 2021 [97] | 2 | 3 | 1 | 2 | 1 | 2 | 2 |
| Sadeghi et al., 2021 [98] | 3 | 3 | 3 | 2 | 1 | 2 | 3 |
| Yamagata et al., 2021 [96] | 3 | 3 | 3 | 1 | 1 | 1 | 3 |
| Bruijn et al., 2022 [112] | 3 | 3 | 3 | 1 | 1 | 2 | 3 |
| Figueiredo et al., 2022 [99] | 3 | 3 | 1 | 2 | 1 | 2 | 3 |
| Nascimento et al., 2022 [100] | 2 | 3 | 1 | 2 | 3 | 2 | 2 |
| Wang et al., 2022 [113] | 1 | 3 | 3 | 1 | 1 | 2 | 3 |
| Yoshida et al., 2022 [101] | 2 | 3 | 3 | 2 | 1 | 2 | 3 |
| Baba et al., 2023 [102] | 3 | 3 | 3 | 2 | 1 | 2 | 3 |
| Wang & Bhatt, 2023 [114] | 1 | 3 | 3 | 1 | 1 | 2 | 3 |

1–strong methodological quality; 2—moderate methodological quality; 3—weak methodological quality.

### 3.3. Gait Spatiotemporal Parameters

The spatiotemporal parameters analyzed among the studies comprised gait speed; cadence; stride and step length; stride and step time; stride and step width; stance phase; swing phase; single support phase; double support phase; and base of support.

#### 3.3.1. Gait Speed

Gait speed was the parameter most analyzed, namely in 50 studies that compared fallers and non-fallers. Regarding these studies, 29 reported the fallers' gait speed was lower than non-fallers [14,23,25–27,30–32,34,37,39,43,48–50,55–57,60,63,68,75,84,85,91,93,98,100,102]. Although another 17 studies observed lower values of gait speed in fallers, no statistically significant differences were yielded [22,24,28,36,58,60–62,66,76,80–82,86,89,91,94]. No study reported higher values of gait speed in fallers.

Four studies analyzed gait speed variability using linear measures: one used the coefficient of variation and standard deviation [81], while three only used the standard deviation [24,63,68]. Of these studies, three reported higher values in fallers [24,68,81], while one reported no differences between fallers and non-fallers [63].

Three studies analyzed gait speed during induced slips [110–112] and two during induced trips [103,105]. Four studies reported no differences between falls and recoveries [103,110–112], while only one reported that the gait speed of the elderly who fell was higher than the elderly who recovered from an induced trip [105].

#### 3.3.2. Cadence

Cadence was evaluated in 22 studies that compared fallers with non-fallers. Among these, six studies reported the fallers' cadence was lower [25,32,39,45,68,98]. Although another 10 studies observed lower values of cadence in fallers, no statistically significant difference was yielded [30,31,36,55,61,86,89,94,100,102]. No study reported higher values of cadence in fallers.

One study analyzed cadence variability (using the standard deviation) and observed no differences between fallers and non-fallers [68].

### 3.3.3. Stride and Step Length

Stride or step length was analyzed in 39 studies that compared fallers and non-fallers. Nineteen studies reported the fallers' stride or step length was lower [23,25,30,32–35,39,43,51,53,56,67,84,85,91,94,98,100]. Although another 13 studies observed lower values of stride or step length in fallers, no statistically significant difference was yielded [21,22,24,31,36,44,61,63,66,77,80,81,102]. No study reported higher values of stride or step length in fallers.

Stride or step length variability was studied using the coefficient of variation [14,51,77,81] and standard deviation [24,33,63,84,87]. Three studies reported that fallers yielded higher values [14,24,33], while six reported no differences [51,63,77,81,84,87].

Step or stride lengths were also analyzed in studies that induced slips [109,111,113] and trips [103,105,107] during their methodological set-up. Of these, one study reported the stride and step length of the elderly who fell were higher than the elderly who recovered from the induced trip [105], while two studies reported the stride and step length of the elderly who fell were lower than the subjects who recovered from the induced trips [107] and induced slips [113]. The other four studies reported no differences between falls and recoveries [103,109,111].

### 3.3.4. Stride and Step Time

Stride and step time were evaluated in 37 studies that compared fallers and non-fallers. Nine studies reported the fallers' stride or step time was higher [26,34,40,45,57,68,74,85,98]. Although another 10 studies reported higher values of stride or step time in fallers, no statistically significant differences were yielded [22,24,31,36,43,44,51,65,84,86]. Only one study reported lower values of stride time in fallers [47].

Stride and step time variability were analyzed using the coefficient of variation [45,51,56,60,73,79–81,86] and standard deviation [24,28,40,68,79,84,85]. Six studies reported that fallers yielded higher values [28,40,60,68,79,84], while 10 reported no differences [24,45,51,56,60,73,80,81,85,86].

Step or stride time was also analyzed in studies that induced trips during their methodological set-up [105,107]. These two studies reported the step time of the elderly who fell was lower than the elderly who recovered from induced trips.

### 3.3.5. Stride and Step Width

Stride and step width were analyzed in 18 studies that compared fallers and non-fallers. Three studies showed that fallers' stride or step width was higher [21,51,74], while one study observed lower values of step width in fallers [51]. The other studies showed no differences between fallers and non-fallers [20–22,24,53,59,63,77,78,81,87,93,94,98,101].

Stride and step width variability was evaluated using the coefficient of variation [51,77,78,81] and standard deviation [20,24,63,87]. All studies reported no differences between fallers and non-fallers.

Step width was also analyzed in studies that induced slips [108] and trips [105] during their methodological set-up. Both reported no differences between fallers and those who recovered [105]. Step width variability was also evaluated (using standard deviation) in one study [109], which verified higher values in fallers.

### 3.3.6. Stance Phase

The stance phase was evaluated in 13 studies that compared fallers and non-fallers. Four studies reported the fallers' stance phase was higher [40,55,84,85]. The other studies showed no differences between fallers and non-fallers [45,47,51,56,63,73,77,86,102].

The stance phase variability was evaluated using the coefficient of variation [45,51,73,77,79] and standard deviation [40,63,85]. Five studies reported that fallers yielded higher values [40,63,77,79,85], while three reported no differences [45,51,73].

The stance phase was also evaluated in one study that induced trips during its methodological set-up [103], which showed no differences between fallers and those who recovered.

### 3.3.7. Swing Phase

The swing phase was analyzed in 18 studies that compared fallers and non-fallers. Five studies observed the fallers' swing phase was lower than non-fallers [14,31,55,84,85]. In the other 13 studies, no statistically significant difference was yielded [21,22,40,44,45,47, 51,56,63,73,77,86,94].

The swing phase variability was studied using the coefficient of variation [14,45,51,73,77] and standard deviation [28,40,63,79,85]. Three studies reported that fallers yielded higher values [14,28,79], while seven reported no differences [40,45,51,63,73,77,85].

The swing phase was analyzed in one study that induced trips during its methodological set-up [103], which reported no differences between fallers and those who recovered.

### 3.3.8. Single Support Phase

The single support phase was evaluated in 10 studies that compared fallers and non-fallers. The fallers' single support phase was lower than non-fallers in two studies [39,55]. The other eight studies showed no differences between fallers and non-fallers [21,22,31,36, 45,51,80,84].

The single support phase variability was also analyzed in two studies using the coefficient of variation [45,51], which observed no differences between fallers and non-fallers.

### 3.3.9. Double Support Phase

The double support phase was analyzed in 18 studies that compared fallers and non-fallers. Of these, nine studies reported the fallers' double support phase was higher [14,26, 31,33,39,55,84,85,98], while one observed exactly the contrary [45]. The other eight studies showed no difference between fallers and non-fallers [21,24,44,51,56,73,102,103].

The double support phase variability was studied using the standard deviation [24] and coefficient of variation [45,51]. One study reported that fallers yielded higher values [24], while two reported no differences [45,51].

The double support phase was also evaluated in one study that induced trips during its methodological set-up [103], which reported no differences between falls and recoveries.

### 3.3.10. Base of Support during Gait

The base of support during gait was analyzed in four studies that compared fallers and non-fallers. Of these, two studies reported the fallers' base of support was higher [26,56]. The other two studies showed no differences between fallers and non-fallers [22,49].

The margin of dynamic stability was evaluated in one study that compared fallers and non-fallers [74]; the authors found higher values in fallers. This parameter was also analyzed in one study that induced slips during its methodological set-up [109]. In this study, the authors found higher values in fallers.

### 3.3.11. Others Parameters

The time of toe-off occurrence (% of the gait stride) was analyzed in one study that compared fallers and non-fallers [89]. This study reported no differences between fallers and non-fallers regarding this parameter.

### 3.4. Kinematic Parameters

The kinematics parameters analyzed among the studies comprised the following: minimum foot/toe clearance; center of mass (CoM); center of pressure (CoP); head, trunk, pelvis, and lower limb kinematics; and slip kinematic parameters.

### 3.4.1. Minimum Foot/Toe Clearance

The minimum foot/toe clearance was analyzed in nine studies that compared fallers and non-fallers. Two studies reported the fallers' minimum foot/toe clearance was lower [34,66]. Nonetheless, three studies reported contrary results, i.e., fallers' minimum foot/toe clearance was higher [37,42,89]. On the other hand, four studies reported no differences between fallers and non-fallers [21,41,63,91].

The minimum foot/toe clearance variability was studied using linear measures (i.e., coefficient of variation [34,91] and standard deviation [37,63,91]) and nonlinear measures (i.e., approximate entropy [38,41], sample entropy [38], wavelet-based multiscale exponent [42], detrended fluctuation analysis exponent [42], and Poincaré plot indexes [41,42]). Four studies reported that fallers yielded higher variability [34,37,38,41,42], while two studies reported no differences between fallers and non-fallers [63,91].

### 3.4.2. CoM

Differences between fallers and non-fallers regarding CoM position were found in two studies [71,93], while one showed no differences [49]. One study found no differences regarding CoM displacement [96], while another found differences regarding CoM lateral sway [35]. Three studies observed lower values of the fallers' CoM velocity during gait [53,71,74], namely AP velocity [53,74], while one found no differences [105]. Regarding CoM acceleration, one study found lower values in fallers [71].

CoM variability was analyzed in one study using the variance [94]. In this study, higher values were found in fallers regarding the variability in the CoM vertical displacement; however, concerning the variability in the CoM mediolateral (ML) displacement, no differences were observed between fallers and non-fallers. On the other hand, one study using the local divergence exponent found higher values in fallers, i.e., higher variability [77].

CoM was also analyzed in one study that induced trips during its methodological set-up, which reported the CoM position and velocity of the elderly who fell were not different than the elderly who recovered from the induced trip [108].

The dynamic stability of CoM was analyzed in one study that induced slips during its methodological set-up (dynamic stability is the relative motion state between CoM and the base of support). This study found no differences between the elderly who fell and the elderly who recovered from the induced slip [109]. The dynamic stability was also analyzed in three studies that induced slips during their methodological set-up. Of this, one study reported the fallers' dynamic stability was higher [104]. The other two studies showed no differences [110,111].

### 3.4.3. CoP Kinematics

One study reported that the fallers' CoP excursion index was lower than non-fallers [97]. Moreover, one study found higher CoP ML displacement in fallers [101]. On the other hand, four studies reported the CoP AP displacement and/or velocity presented no differences between fallers and non-fallers [73,86,97,101].

The variability in the CoP AP and ML displacements was evaluated in two studies using the standard deviation [81] and coefficient of variation [73]. Their authors reported no differences between fallers and non-fallers.

### 3.4.4. CoM–CoP Relation

Two studies that compared fallers and non-fallers analyzed the CoM–CoP relation. Their results are contradictory. While one study reported the fallers' CoM–CoP AP distance was lower [49], the other observed higher values [69]. On the other hand, CoM–CoP ML distance presented no difference between fallers and non-fallers in one study [49].

### 3.4.5. Head, Trunk, and Pelvis Linear Kinematics

Trunk linear kinematics were evaluated in one study that compared fallers and non-fallers [34]; they found higher maximal ML displacement of the trunk center in fallers.

One study used the refined composite multiscale entropy and the refined multiscale permutation entropy regarding lower back velocity and acceleration [72]; they found higher complexity in fallers. The computed multiscale entropy and the Shannon entropy were also used to analyze the complexity of the trunk AP and ML displacement [76]; data pointed out the inability of the multiscale entropy to distinguish fallers and non-fallers, whereas Shannon entropy seemed to be sufficient in fall risk prediction. On the other hand, fallers presented higher variability in the lower back vertical axis and lower variability in the lower back ML axis [57].

One study used the short-term exponents of the trunk ML, AP, and vertical displacement to analyze gait variability. No differences were yielded between fallers and non-fallers [95]. One study that during their methodological set-up induced slips [109] also analyzed the variability in the trunk through nonlinear measures, i.e., using the maximum Lyapunov exponent and Floquet multiplier. Their authors found no differences between fallers and those who recovered.

The maximum Lyapunov exponent was also used in two studies in order to evaluate the gait variability. Contradictory results were found in these two studies, i.e., one found higher variability in the right anterior superior iliac spine in fallers [43], while the other did not find differences between fallers and non-fallers regarding the head and pelvis [73]. One of these studies also analyzed variability using the ratio of even to odd harmonics, having found differences between fallers and non-fallers regarding the pelvis [73].

### 3.4.6. Lower Limb Linear Kinematics

Two studies analyzed the foot velocity and heel vertical velocity at heel strike [44,66]. They found no differences between fallers and non-fallers.

One study used the fast Fourier transform first quartile on the shank displacement [86]; they found higher variability in fallers.

The hip horizontal displacement [103], the hip–ankle distance, the obstacle–ankle distance, the ankle horizontal velocity, and the hip vertical velocity [107] were analyzed in two studies that induced trips during their methodological set-up. No differences were found between fallers and those who recovered.

The hip height was analyzed in two studies that induced trips during their methodological set-up [103,107]. Of this, one study reported the fallers' hip height at ground foot contact was lower [107]. The other study showed no differences between the elderly who fell or recovered from the induced trips [103].

### 3.4.7. Slip Kinematics Parameters

The slip distance was analyzed in three studies that induced slips during their methodological set-up [110,111,113]. Of these, two studies reported the fallers' slip distance was higher [110,113]. The other study showed no differences [111]. The peak slip velocity was also evaluated in these three studies. Of these, one study reported the fallers' peak slip velocity was higher [113]; the other two studies showed no differences [110,111]. One of these three studies also assessed the slip duration [111]. In this study, no differences were found between fallers and recoverers.

### 3.5. Angular Kinematic Parameters

The angular kinematics analyzed among the selected studies comprised the lower limb joints (hip, knee, and ankle), foot progression angle, and foot angle with the ground, trunk, pelvis, thigh, and shank.

### 3.5.1. Hip

Five studies reported differences between fallers and non-fallers regarding hip angular position or displacement [29,31,35,58,63]. On the other hand, eight studies found no differences regarding hip angular position or displacement [21,22,29,31,61,82,98,102].

Fallers exhibited greater variability in the hip in the frontal plane during the entire stance phase [63].

Hip angular position was also analyzed in two studies that induced trips [107] and slips [111] during their methodological set-up. The first one found differences between fallers and those who recovered, while the other did not.

### 3.5.2. Knee

Knee kinematics were evaluated in 10 studies that compared fallers and non-fallers. These studies reported no differences regarding knee angular position or displacement [21, 22,30,31,61,82,83,90,98,102].

Fallers exhibited greater variability in the knee during the entire swing phase [63].

Knee angular position was also analyzed in one study that induced slips during its methodological set-up [111]. Data yielded higher values of knee flexion in fallers but no differences regarding knee extension.

### 3.5.3. Ankle

Ankle angular position or displacement yielded no differences between fallers and non-fallers in nine studies [21,22,30,58,61,82,90,98,102]. On the other hand, differences between fallers and non-fallers regarding ankle kinematics were found in three studies [31,35,63].

Fallers exhibited greater variability in the ankle in the frontal plane during the entire stance phase [63].

Ankle angular position was analyzed in one study that induced slips during its methodological set-up [111]. No differences were found between fallers and non-fallers.

### 3.5.4. Foot Progression Angle

Foot progression angle was analyzed in three studies that compared fallers and non-fallers. These studies supported no differences between fallers and non-fallers [84,90,94].

Foot progression angle variability was also studied using the standard deviation [93,94]. No differences were found between fallers and non-fallers.

### 3.5.5. Foot Angle with Ground

Differences between fallers and non-fallers regarding the foot angle with the ground were found in two studies [34,102]. While one study observed higher values in fallers [34], the other found lower values [102].

The variability in the maximum foot angle with the ground was also studied using the coefficient of variation [34]. In this study, higher values were found in fallers.

### 3.5.6. Trunk

Trunk angular position was analyzed in one study that compared fallers and non-fallers; no differences were found [54].

Trunk angular position was also evaluated in three studies that induced trips [105,107] and slips [114] during their methodological set-up. Two of these studies observed differences between fallers and non-fallers [107,114], while the other did not [105].

### 3.5.7. Pelvis

Pelvis angular position was analyzed in three studies that compared fallers and non-fallers. All studies reported no differences between fallers and non-fallers [30,93,94].

The variability in pelvis angular position was studied using the standard deviation [93,94]; however, no differences were yielded between fallers and non-fallers.

### 3.5.8. Thigh

Thigh angular position was analyzed in two studies that compared fallers and non-fallers [93,94], which reported no differences between these groups.

The variability in thigh angular position was studied using the standard deviation [93,94]. No differences between fallers and non-fallers were found.

### 3.5.9. Shank

Three studies that compared fallers and non-fallers reported differences between fallers and non-fallers regarding shank angular position [45,93,94].

The variability in the shank angular position was studied using the standard deviation [93,94]. Differences between fallers and non-fallers were found, with higher values in fallers.

Shank angular position was also analyzed in two studies that induced slips during their methodological set-up [110,111]. No differences between fallers and non-fallers were yielded.

### 3.5.10. Other Parameters

AP CoM–ankle inclination was evaluated in one study that compared fallers and non-fallers [53]; the authors found higher inclinations in fallers.

The ankle–hip inclination at the time of loading was analyzed in one study that induced trips during its methodological set-up [107]. In this study, differences between fallers and those who recovered were found, i.e., higher inclinations in fallers.

Two studies analyzed the variability in inter-joint coordination. One study used the standard deviation for this purpose [55] and found higher variability in fallers regarding the knee–ankle coordination (during stance and swing phase); however, the variability in hip–knee coordination yielded no differences between fallers and non-fallers [55]. In addition, another study reported lower variability in the lower limb coordination in fallers, indicating an inconsistency in neuromuscular control [98].

### *3.6. Kinetic Parameters*

### 3.6.1. Ground Reaction Force

The ground reaction force was analyzed in five studies that compared fallers and non-fallers [54,69,73,84,86]. The peak braking force and the peak propulsive force (calculated from the AP component) presented no differences between fallers and non-fallers [69,84]. On the other hand, differences regarding braking force and propulsive force were found in another study [54]. On the other hand, the impulse during the gait cycle presented no differences in two studies [73,86].

Ground reaction force was also evaluated in one study that induced slips during its methodological set-up [101]. This study reported the fallers' coefficient of friction (horizontal ground reaction force/vertical ground reaction force after heel contact) was lower than in those who recovered.

### 3.6.2. Plantar Pressure

Peak plantar pressure and total pressure–time integral presented higher values in fallers [46,97]. On the other hand, the pressure–time integral regarding different foot regions (medial and lateral heel, medial and lateral midfoot, central and lateral forefoot, hallux, second and lateral toe) showed no difference between fallers and non-fallers; only the pressure–time integral of medial forefoot yielded higher values in fallers [98].

### *3.7. Dynamic Parameters*

### 3.7.1. Hip Moment

The hip moment was analyzed in four studies that compared fallers and non-fallers [25,30,31,54]. Three studies point out differences, namely in the sagittal plane [25,30,31]. Only one study reported no differences between fallers and non-fallers [54]. Moreover, one study also evaluated other planes of movement [30]; they found differences regarding the hip adduction moment but no differences concerning hip abduction, external, and internal moments.

The hip moment was analyzed in one study that induced trips during its methodological set-up [103], which reported no differences between fallers and those who recovered.

### 3.7.2. Knee Moment

The knee moment was analyzed in four studies that compared fallers and non-fallers. Two studies yielded differences between fallers and non-fallers [25,30], while the other two reported no differences [31,54].

The knee moment was also evaluated in one study that induced trips during its methodological set-up [103]. In this study, no differences were yielded.

### 3.7.3. Ankle Moment

The ankle moment was evaluated in four studies that compared fallers and non-fallers. Two studies yielded differences between fallers and non-fallers [25,54], while the other two reported no differences [30,31].

Ankle moment was also analyzed in one study that induced trips during its methodological set-up [103]. The authors reported no differences between fallers and non-fallers.

### 3.7.4. Hip, Knee, and Ankle Power Absorption and Generation

One study found differences between fallers and non-fallers regarding hip power absorption and generation, namely with lower values in fallers [30]. However, another two studies reported no differences between fallers and non-fallers [25,31].

Two studies reported the knee power absorption of fallers was lower than non-fallers; however, the knee power peak and power generation presented no differences [25,30].

Ankle power absorption and generation were analyzed in three studies that compared fallers and non-fallers [25,30,31]. One study reported the ankle power generation of fallers was lower than non-fallers [30], while another observed higher values of ankle power absorption in fallers [25]. On the other hand, another study reported the ankle power peak presented no differences between fallers and non-fallers [31].

### 3.7.5. Other Parameters

One study found differences between fallers and non-fallers regarding total CoM kinetic energy (at swing-off), namely with lower values in fallers [53].

Angular momentum was also evaluated in one study that induced trips during its methodological set-up [103]. This study reported the fallers' angular momentum at push-off and total momentum are predictors of falls.

### *3.8. EMG Parameters*

### 3.8.1. Muscle Activity

Muscle activity was analyzed in 10 studies using EMG measures. Eight studies compared elderly fallers and non-fallers [47,59,70,82,88,89,92,94], while two studies compared the elderly who fell with the elderly who recovered from induced slips [110,111].

The fallers' internal oblique activity before heel contact was lower than non-fallers. On the other hand, the same study observed no significant differences in internal oblique activity at the initial stance, final stance, and after toe-off [59].

The gluteus maximus was analyzed in one study that compared fallers and non-fallers. This study reported the fallers' gluteus maximus activity at the final stance is higher than non-fallers. On the other hand, no significant differences in gluteus maximus activity were observed at the initial stance, before heel contact, and after toe-off [59].

The fallers' biceps femoris activity at initial stance and before heel contact is higher than non-fallers. On the other hand, no significant differences in biceps femoris activity were observed at the final stance and after toe-off [59].

The biceps femoris long head was analyzed in one study that induced slips during its methodological set-up. Their authors pointed out the higher onset latencies of fallers. On

the other hand, no differences were observed regarding the onset latencies of the nonslip leg and the peak magnitude of the slip and nonslip leg [110].

The gastrocnemius was analyzed in two studies that compared fallers and non-fallers [47,59]. One of these studies reported the fallers' gastrocnemius activity during the stance phase was lower than non-fallers [47]. On the other hand, no differences were observed in gastrocnemius activity at the initial stance, before heel contact, and after toe-off [47,59].

The medial gastrocnemius yielded differences in onset latencies and peak magnitude when the elderly who fell and the elderly who recovered from an induced slip were compared [110].

The vastus lateralis was analyzed in one study that induced slips during its methodological set-up. The authors reported the fallers' onset latencies of the slip leg were higher than in those who recovered. On the other hand, no differences in onset latencies of the nonslip leg and peak magnitude of the slip and nonslip leg were reported [110].

No differences between fallers and non-fallers were found regarding soleus activity and onset latency [47], tibialis anterior activity [59,110], latency [47,110], peak magnitude [110], multifidus activity [59,110], or rectus femoris activity [59,110].

The variability in central locus activation of the rectus femoris was studied using the coefficient variation [98]. No differences between fallers and non-fallers were found.

### 3.8.2. Muscle Synergies and Co-Contraction

The co-contraction index (between tibialis anterior and gastrocnemius) and lower limb muscle co-contractions of fallers were higher than non-fallers [59,70].

The muscle synergies were analyzed in two studies that compared fallers and non-fallers. One study reported the fallers' muscle synergies were lower than non-fallers [82]. On the other hand, fallers' kinematic synergy during the early and late swings was higher than non-fallers [94].

The muscle synergies were also evaluated in two studies that induced slips during their methodological set-up. In both studies, it was reported that the fallers' muscle synergies were lower than in those who recovered [110,111].

### 3.9. Gait Symmetry and Gait Smoothness

Two studies found differences between fallers and non-fallers regarding gait symmetry, which expressed the similarity of craniocaudal movements on the left and the right independently from fluctuations in successive craniocaudal movements of each limb [32,68].

The two studies that analyzed gait smoothness—a quality that reflects the continuousness or non-intermittency of walking [68,99]—found different results, i.e., one study reported that gait smoothness was associated with the number of falls [68], while the other did not find differences between fallers and non-fallers [99].

### 4. Discussion

The aim of the present study was to conduct a systematic review to identify and describe the gait biomechanical parameters related to falls in the elderly population. According to the results of this systematic review, the gait spatiotemporal parameters were the most analyzed data when elderly fallers and non-fallers were compared, especially the gait speed. The majority of the selected studies for this systematic review reported lower gait speed in elderly fallers, pointing out that this can be a gait biomechanical parameter that differentiates elderly fallers from non-fallers. This lower gait speed in fallers may be the result of reduced functional capacity, fear of falling, or both. Concerning functional capacity, the data compiled in this systematic review may provide some clues: differences between elderly fallers and non-fallers were found regarding lower limb muscular activity and lower limb joints biomechanics (i.e., joints moments and powers), although the number of studies on these topics has been scarce. On the other hand, gait speed has been associated with a fear of falling [24,52,60]. In this way, 63.4% of the retrospective studies (i.e., studies

in which fallers had already suffered a fall at the time of gait assessment) that analyzed gait speed reported a lower gait speed in fallers, while only 33.3% of the prospective studies (i.e., studies in which fallers had not suffered a fall at the time of gait assessment) reported a lower gait speed in fallers. These numbers also suggest an effect of fear of falling again on the elderly gait, namely on gait speed. Other parameters were also referenced as related to the fear of falling, such as the stride/step length and the double support phase [24]. According to the data in this systematic review, these parameters have also presented the ability to differentiate elderly fallers from non-fallers, i.e., fallers tended to present a reduced stride/step length and an increased double support phase. Therefore, these data point to the importance of interventions in the elderly who restore and improve their functional capacity and self-confidence during activities of daily living.

Gait speed is dependent on cadence and step length [115]. According to data from this systematic review, the lower gait speed shown by fallers seems to be more sensitive to a reduction in step length than in cadence, although several studies have also found a lower cadence in fallers. On the other hand, stride and step time were other spatiotemporal parameters associated with gait speed [115]; according to our data, these parameters showed lower ability than gait speed to differentiate fallers from non-fallers. Finally, stride/step width and foot progression angle were gait biomechanical parameters that clearly did not differentiate fallers from non-fallers.

Tripping is one of the most frequent causes of falls in the elderly [9], while the minimum foot/toe clearance has been a gait biomechanical parameter associated with trips [11]. Regarding this parameter, our data point to contradictory and non-differentiating results, suggesting the minimum foot/toe clearance may not be a consistent differentiator between fallers and non-fallers. In contrast, the minimum foot/toe clearance variability, using both linear and nonlinear measures, appears as a potential parameter associated with a history of falls in the elderly. This is in line with a previous systematic review [11], which concluded that higher minimum foot clearance variability may contribute to an increased risk of trips.

The minimum foot/toe clearance Is sensitive to alterations in the angular positions of the swing limb joints, i.e., hip, knee, and ankle [116]. Previous research pointed to a higher sensitivity to the ankle angular position, then to the hip angular position, and lastly to the knee angular position [116–118]. Thus, the elderly can adjust minimum foot/toe clearance by controlling these joint angles. In this way, the analysis of lower limb joint kinematics among fallers and non-fallers is an important issue. Although it was not transversal to all studies selected for this systematic review, some of them identified differences between fallers and non-fallers regarding hip and ankle angular position or displacement. Thus, the need for interventions that improve the motor control of the lower limb joints is a very important aspect to explore. In this regard, a program of proprioceptive and functional strength exercises seems to be a good solution for improving motor control of the lower limb joints [119,120]. On the other hand, knee kinematics do not seem to have the ability to differentiate fallers from non-fallers.

Studies involving gait with induced stability perturbations were relatively scarce and analyzed a large disparity of variables (with low frequency in the various studies). Among the studies that induced trips during their methodological set-up, a lowering strategy, characterized by a faster gait speed and delayed support limb loading, was linked to falls during a step [107]. On the other hand, elevating strategy falls were marked by an accelerated gait speed and excessive lumbar flexion. Moreover, fallers exhibited an insufficient reduction in angular momentum during push-off, improper recovery limb placement, and reduced rates of moment generation in support limb joints. Due to the fact that research on this issue is scarce, the extraction of conclusions is largely limited.

Slip falls are a growing health concern for the elderly [113]. Heel velocity and foot angle with the ground at heel strike were gait biomechanical parameters associated with slip falls [12,34,102]. According to data from this systematic review, the literature is scarce regarding these parameters. Moreover, foot angle with the ground at the heel strike revealed contradictory results, and heel velocity at the heel strike showed no differences between

fallers and non-fallers. Moreover, the studies that induced slips during their methodological set-up assessed other gait biomechanical parameters, such as the coefficient of friction, the slip distance, the peak slip velocity, and the slip duration. Once again, the studies that analyzed these issues are scarce; nonetheless, their data indicated the fallers' coefficient of friction (horizontal ground reaction force/vertical ground reaction force, after heel contact) was lower than in those who recovered. Regarding the other parameters, there seems to be no great ability to differentiate fallers from those who recovered.

Postural stability can be defined as the ability to maintain adequate sustainability of the body along the movement [121]. The CoM and CoP have been used to analyze postural stability in previous studies [122–124]. The literature is scarce concerning the relation between falls and CoM and CoP; however, the contradictory findings across studies emphasize the complexity of assessing postural control and stability during gait, especially regarding the comparison between fallers and non-fallers. Moreover, the analysis of the base of support during gait as well as the margin of dynamic stability also revealed diverse findings, making it impossible to draw clear conclusions.

The variability in the gait biomechanical parameters was studied using linear and nonlinear measures. Overall, fallers tend to exhibit higher variability in the gait pattern. According to previous research [24], this higher gait variability can be linked with lower motor control. As described in a previous paragraph, exercise programs can be good options in order to improve motor control and, as a result, to reduce the risk of falls.

The majority of the included studies were classified as weak in the global assessment, reflecting concerns about their quality. Selection bias was a domain that influenced this classification, with the majority of studies categorized as weak. The dependence on convenience samples has raised concerns regarding the generalizability to broader populations, restricting the external validity of these studies. Additionally, all studies were rated as weak in the study design domain, predominantly attributable to the cross-sectional nature of the investigations and the nonrandomized selection of the subjects. This is not precisely a problem, as our study aimed to compare the gait biomechanical parameters between elderly fallers and non-fallers. Other limitations identified across the various studies include the definition of a fall, the timeframe considered for the fall's occurrence, and the gait assessment on a treadmill. Approximately 55% of the selected studies presented a definition of fall and these definitions were quite similar. However, nearly half of the studies did not provide any explicit definition, which may impact the reliability of the reported results. According to data from this systematic review, approximately 21% of the selected studies conducted the gait assessment on a treadmill. Walking on the treadmill does not reflect everyday gait as it imposes a different gait pattern [125]. In this way, the obtained results may be influenced by this methodological constraint.

Practical and clinical implications arise from this study. In this way, healthcare workers and clinicians must pay attention to some gait biomechanical parameters (i.e., gait speed, stride/step length, and double support phase) when evaluating elderly gait and during interventions that aim to prevent the occurrence of falls.

Some of the gait biomechanical analysis methods used in the selected studies are not considered the gold standard methods to assess gait, i.e., optoelectronic gait analysis systems and force plates [126–128]. Nonetheless, most of the other equipment used in the various studies is presented in the literature as validated and reliable. This heterogeneity observed between the reviewed studies regarding the gait biomechanical analysis methods used may limit our conclusions, contributing to different values for the same parameter. Therefore, this is one of the reasons why no meta-analysis was carried out. Therefore, it is preferable for future investigations to use gold-standard methods to assess gait in the elderly.

Indeed, the scarce literature regarding some parameters limited the ability of this study to yield strong conclusions. In this way, further work is needed to understand the association between the gait biomechanical parameters and falls in the elderly. An example is the need for research that comprises parameters associated with motor control, such as

muscular activity. Another aspect that we did not see addressed in the studies selected for this systematic review was joint stability. One parameter used to study joint stability is dynamic joint stiffness [121], which has been used to differentiate fallers and non-fallers in certain clinical populations [129].

## 5. Conclusions

The results of this systematic review pointed out that the gait speed, stride/step length, and double support phase are biomechanical parameters of gait that play a distinctive role in differentiating fallers from non-fallers. In this way, these are parameters that healthcare workers and clinicians must pay attention to when evaluating elderly gait and during interventions that aim to prevent the occurrence of falls. Elderly fallers also tend to exhibit higher variability; the variability in the minimum foot/toe clearance is an important example due to its relation with trips. Although studies on lower limb muscular activity and joint biomechanics are limited, the available research indicated that differences in these aspects may also be associated with the propensity for falls. However, it is crucial to highlight the complexity of drawing clear conclusions due to the scarcity of literature and contradictory results among studies, namely parameters related to postural stability. Parameters such as minimum foot/toe clearance, step width, and knee kinematics did not demonstrate a discriminative ability between fallers and non-fallers. Therefore, despite advancements in understanding the biomechanics of gait concerning falls, further research is needed at some points to provide a more comprehensive and consistent understanding of these complex relationships.

**Author Contributions:** Conceptualization, J.A., T.A. and P.A.; methodology, J.S., T.A. and P.A.; validation, J.A., T.A. and P.A.; formal analysis, J.A., T.A. and P.A.; investigation, J.S. and P.A.; resources, J.S. and P.A.; data curation, J.S. and P.A.; writing—original draft preparation, J.S.; writing—review and editing, J.A., T.A. and P.A.; supervision, P.A.; project administration, P.A. All authors have read and agreed to the published version of the manuscript.

**Funding:** This research received no external funding.

**Institutional Review Board Statement:** Not applicable.

**Informed Consent Statement:** Not applicable.

**Data Availability Statement:** Not applicable.

**Conflicts of Interest:** The authors declare no conflicts of interest.

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
