# Peer review of "Gait Biomechanical Parameters Related to Falls in the Elderly: A Systematic Review"

_2673-7078, doi:10.3390/biomechanics4010011_

Round 1

Reviewer 1 Report

Comments and Suggestions for Authors

The manuscript provides a systematic review focused on identifying and describing gait biomechanical parameters associated with falls in the elderly population. The research question addressed is the association between specific gait biomechanical parameters and falls in elderly individuals. The methodology employed a systematic review approach, adhering to the Preferred Items for Reporting for Systematic reviews and Meta-Analysis (PRISMA) statement. It also utilized the Quality Assessment Tool for Quantitative Studies to evaluate the methodological quality of the included studies.

Key findings reveal that gait speed, stride/step length, and double support phase are distinct biomechanical parameters of gait that differentiate fallers from non-fallers. The review underscores the necessity for further research to explore additional parameters associated with motor control, muscular activity, and joint stability to better understand the link between gait biomechanical parameters and falls in the elderly.

While the article effectively presents the reviewed studies' findings concisely, it could enhance its analysis by delving into the causal relationships between biomechanical parameters of gait in older individuals experiencing falls versus those who do not. This deeper exploration could elucidate correlated gait parameters likely contributing to falls.

Another limitation of the study lies in the significant heterogeneity observed among the reviewed studies regarding the biomechanical gait analysis methods employed. This diversity leads to the aggregation of variables (such as stride length) based solely on their magnitude, irrespective of the measurement method used. For instance, measurements may involve footprints using ink, electronic footprints on a piezosensor treadmill, a triaxial accelerometer on the body, or an optoelectronic gait analysis system.

In my view, this methodological heterogeneity contributes to variations in the accuracy and reliability of reported values for the same parameter. Consequently, it is not entirely appropriate to amalgamate findings from disparate studies examining the same biomechanical parameter.

While it is valuable to include all relevant studies regardless of the age of the research methods used, I believe the comparative analysis of findings for each parameter should be restricted to studies employing optoelectronic gait analysis systems, either exclusively or in conjunction with electromyographs or plantar pressure platforms. These systems presumably offer a higher level of consistency in measurements. Other studies should be excluded from the sample, and measurement methods other than optoelectronic systems should be added to the exclusion criteria.

Implementing this approach would facilitate a more nuanced exploration of the mutual causal relationships between measured variables, thereby allowing for the development of a model depicting correlated gait variables likely to contribute to falls in older individuals.

Furthermore, while appropriately acknowledging studies involving gait with induced stability perturbations, the review could enhance its analysis by describing correlated gait variables likely to contribute to falls in these scenarios. This could be accomplished by dedicating a separate section of the discussion to explore the nuances of these variables and their implications for falls prevention and intervention strategies.

In conclusion, while the manuscript effectively highlights key findings regarding gait biomechanical parameters and falls in the elderly, addressing the mentioned areas for improvement could enhance the depth and robustness of its analysis.

Author Response

Dear reviewer.

Thank you very much for revisions and suggestions. We truly believe that this revisions and suggestions will improve the manuscript.

The manuscript provides a systematic review focused on identifying and describing gait biomechanical parameters associated with falls in the elderly population. The research question addressed is the association between specific gait biomechanical parameters and falls in elderly individuals. The methodology employed a systematic review approach, adhering to the Preferred Items for Reporting for Systematic reviews and Meta-Analysis (PRISMA) statement. It also utilized the Quality Assessment Tool for Quantitative Studies to evaluate the methodological quality of the included studies.

Key findings reveal that gait speed, stride/step length, and double support phase are distinct biomechanical parameters of gait that differentiate fallers from non-fallers. The review underscores the necessity for further research to explore additional parameters associated with motor control, muscular activity, and joint stability to better understand the link between gait biomechanical parameters and falls in the elderly.

While the article effectively presents the reviewed studies' findings concisely, it could enhance its analysis by delving into the causal relationships between biomechanical parameters of gait in older individuals experiencing falls versus those who do not. This deeper exploration could elucidate correlated gait parameters likely contributing to falls.

Another limitation of the study lies in the significant heterogeneity observed among the reviewed studies regarding the biomechanical gait analysis methods employed. This diversity leads to the aggregation of variables (such as stride length) based solely on their magnitude, irrespective of the measurement method used. For instance, measurements may involve footprints using ink, electronic footprints on a piezosensor treadmill, a triaxial accelerometer on the body, or an optoelectronic gait analysis system.

In my view, this methodological heterogeneity contributes to variations in the accuracy and reliability of reported values for the same parameter. Consequently, it is not entirely appropriate to amalgamate findings from disparate studies examining the same biomechanical parameter.

While it is valuable to include all relevant studies regardless of the age of the research methods used, I believe the comparative analysis of findings for each parameter should be restricted to studies employing optoelectronic gait analysis systems, either exclusively or in conjunction with electromyographs or plantar pressure platforms. These systems presumably offer a higher level of consistency in measurements. Other studies should be excluded from the sample, and measurement methods other than optoelectronic systems should be added to the exclusion criteria.

Implementing this approach would facilitate a more nuanced exploration of the mutual causal relationships between measured variables, thereby allowing for the development of a model depicting correlated gait variables likely to contribute to falls in older individuals.

RESPONSE: Thank you for your comment. We understand the issue raised regarding the disparity in methodologies used in the various studies selected for this systematic review. It is well documented that optoelectronic gait analysis systems, as well as force plates, are considered the gold standard equipment to measure gait parameters. Nonetheless, most of the others equipment used in the various studies are also presented in the literature as validated and reliable. An example of this is the inertial sensor systems, which are clearly validated and reliable. The few studies that used less valid methodologies or equipment were classified as weak in the Data Collection Methods component (6 studies; please see Table 3).

If the aim of the present study were to conduct a meta-analysis, we would have no doubts about carrying out it including similar methodologies as inclusion criteria. And then, your suggestion would have our full agreement. Moreover, in the Discussion section we discuss the practical and clinical implications arise from this study, i.e., healthcare and clinicians must pay attention to some gait biomechanical parameters (i.e., gait speed, stride/step length, and double support phase) when evaluating elderly gait and during interventions that aim to prevent the occurrence of falls. If we only included articles that used gold standard methodologies we would not be able to draw any implications for clinical practice, as in most realities this equipment does not exist. In this way, we decided to recognize the issue you raised as a limitation, i.e., the following paragraph was added to Discussion section: “Some of the gait biomechanical analysis methods used in the selected studies are not considered the gold standard methods to assess gait, i.e., optoelectronic gait analysis systems and force plates [127–129]. Nonetheless, most of the other equipment used in the various studies is presented in the literature as validated and reliable. This heterogeneity observed between the reviewed studies regarding the gait biomechanical analysis methods used may limit our conclusions, contributing to different values for the same parameter. Therefore, this is one of the reasons why no meta-analysis was carried out. Therefore, it is preferable for future investigations to use gold standard methods to assess gait in the elderly.”

Furthermore, while appropriately acknowledging studies involving gait with induced stability perturbations, the review could enhance its analysis by describing correlated gait variables likely to contribute to falls in these scenarios. This could be accomplished by dedicating a separate section of the discussion to explore the nuances of these variables and their implications for falls prevention and intervention strategies.

RESPONSE: Thank you for your comment. Indeed, we tried to do that. We have a paragraph in Discussion section (5th) that addresses the issue of trip-inducing studies, and we have the next paragraph that addresses the issue of slip-inducing studies. The problem is that the literature in this field is relatively scarce and evaluates a large disparity of variables with low frequency in the various studies. In this way, we believe that strong conclusions are inappropriate on these two points. However, some text has been added to the 5th paragraph to highlight this point: “Studies involving gait with induced stability perturbations were relatively scarce and analyzed a large disparity of variables (with low frequency in the various studies). (…) Due to the fact that research on this issue is scarce, the extraction of conclusions is largely limited.

Reviewer 2 Report

Comments and Suggestions for Authors

My review is as below;

1. The search had been identified on 13988 studies. The number of searches are nice. From these, 96 were selected is sufficient.  Gait speed, stride/step length, and double support phase had been determined gait biomechanical parameters that differentiated fallers from non-fallers were valuable. In addition to this although the studies were scarce, differences between fallers and non-fallers had been found regarding lower limb muscular activity and less joint biomechanics may be acceptable.

2.Indeed because of less literature  number  and contradictory results among studies had been not clear and  minimum foot/toe clearance, step width, and knee kinematics did not differentiate fallers from non-fallers are weakness of the search.

3. Both 'Table 1. Characteristics and data of the studies that compared fallers and non-fallers gait.' and 'Table 2. Characteristics and data of the studies that compared elderly who fell during induced falls and elderly who did not.' need abbreviations. Abbreviations can be made especially in gait parameters and universal terms.

4. Can it be added to these tables whether orthosis or insole is used?

Author Response

Dear reviewer.

Thank you very much for revisions and suggestions. We truly believe that this revisions and suggestions will improve the manuscript.

  1. The search had been identified on 13988 studies. The number of searches are nice. From these, 96 were selected is sufficient.  Gait speed, stride/step length, and double support phase had been determined gait biomechanical parameters that differentiated fallers from non-fallers were valuable. In addition to this although the studies were scarce, differences between fallers and non-fallers had been found regarding lower limb muscular activity and less joint biomechanics may be acceptable.

RESPONSE: Thank you for your comment.

2.Indeed because of less literature  number  and contradictory results among studies had been not clear and  minimum foot/toe clearance, step width, and knee kinematics did not differentiate fallers from non-fallers are weakness of the search.

RESPONSE: Thank you for your comment. In fact, the scarce literature regarding some parameters limited the ability of this study to yield strong conclusions. Therefore, the following changes were made to the Discussion section in order to highlight this important point: “On the other hand, the scarce literature regarding some parameters limited the ability of this study to yield strong conclusions. In this way…”

  1. Both 'Table 1. Characteristics and data of the studies that compared fallers and non-fallers gait.' and 'Table 2. Characteristics and data of the studies that compared elderly who fell during induced falls and elderly who did not.' need abbreviations. Abbreviations can be made especially in gait parameters and universal terms.

RESPONSE: Thank you for your comment. These tables already contained some abbreviations, namely: center of pressure - CoP; center of mass - CoM. However, the flaws have been improved. Moreover, the following abbreviations were introduced in Table 1 and 2: minimum toe clearance was abbreviated to MTC; anteroposterior was abbreviated to AP; mediolateral was abbreviated to ML.

  1. Can it be added to these tables whether orthosis or insole is used?

RESPONSE: Thank you for your comment. The tables already describe the use of insoles to collect gait data (i.e., references 28, 73, and 86).  The tables described all inclusion and exclusion criteria of the selected studies. No additional information regarding orthosis and insoles was described in the selected studies.

Reviewer 3 Report

Comments and Suggestions for Authors

Authors presented a review study, which systematically included the biomechanical parameters may relate to the falling or falling risks in elder adults. The topic is interesting and of clinical significance, while several issues should be properly addressed.

In the Introduction, the 2nd paragraph, both intrinsic and extrinsic factors were reported, while the information was not properly organized and presented to justify the aim of this study. More elaboration on the difference between trip falling and slip falling are expected.

Discussion and Conclusion, are there any practical and clinical implications shall be noted for healthcare and clinicians? 

Comments on the Quality of English Language

Quality of English is fine. 

Author Response

Dear reviewer.

Thank you very much for revisions and suggestions. We truly believe that this revisions and suggestions will improve the manuscript.

Authors presented a review study, which systematically included the biomechanical parameters may relate to the falling or falling risks in elder adults. The topic is interesting and of clinical significance, while several issues should be properly addressed.

RESPONSE: Thank you for your comment.

In the Introduction, the 2nd paragraph, both intrinsic and extrinsic factors were reported, while the information was not properly organized and presented to justify the aim of this study. More elaboration on the difference between trip falling and slip falling are expected.

RESPONSE: Thank you for your comment. The second paragraph was changed: “Falls in elderly are dependent on complex multifactorial risks such as: (1) intrinsic risks – e.g., muscle weakness, stability disorders, functional and cognitive impairment, and visual deficits; (2) extrinsic risks – e.g., prescription of four or more medications; (3) environmental risks – e.g., poor lighting or rugs that slide [7]. Focusing our attention on intrinsic factors, it is important to highlight that the subjects’ functional capacity and motor control play an important role in falls. This is particularly important during walking, one of the activities of daily life in which falls are most prevalent [8]. Moreover, the increment of knowledge about gait biomechanics related to falls (i.e., objective data of functional capacity and motor control) may help in the identification of subjects that present high risk of fall and also in the development of interventions to de-crease this risk [9]. In this context, tripping and slipping are the most frequent causes of falls during gait [10]. Regarding tripping, the minimum toe clearance emerged as one gait biomechanical parameter related to this kind of falls. A previous systematic review [11] addressed this issue and found no differences between fallers and non-fallers regarding minimum toe clearance, although the literature found was scarce. Furthermore, this systematic review also found that fallers had greater variability of the minimum toe clearance. Naturally, the biomechanical parameters related to slipping are different. In this way, the heel anteroposterior velocity at heel strike has been a biomechanical gait parameter related to slips [12]. Finally, the gait spatiotemporal parameters were also associated with the falls history, namely: gait speed and cadence [13,14]; stride length, double support phase duration, and variability of the stride length and swing time [14]; variability of the step length and double-support phase [13].”

Discussion and Conclusion, are there any practical and clinical implications shall be noted for healthcare and clinicians?

RESPONSE: Thank you for your comment. We believe the practical and clinical implications have already been addressed during Discussion section:

1st paragraph – “Therefore, these data point to the importance of interventions in the elderly that restore and improve their functional capacity and self-confidence during activities of daily living.”

4th paragraph – “Thus, the need for interventions that improve the motor control of the lower limbs joints is a very important aspect to explore. In this regard, a program of proprioceptive and functional strength exercises seems to be a good solution for improving motor control of the lower limbs joints [119,120].”

8th paragraph – “As described in a previous paragraph, exercise programs can be good options in order to improve motor control, and as result, to reduce the risk of falls.”

Nonetheless, some information was added to Discussion and Conclusions sections.

In Discussion section the following paragraph was added: “Practical and clinical implications arise from this study. In this way, healthcare and clinicians must pay attention to some gait biomechanical parameters (i.e., gait speed, stride/step length, and double support phase) when evaluating elderly gait and during interventions that aim to prevent the occurrence of falls.”

In Conclusions section the following sentence was added: “In this way, these are parameters that healthcare and clinicians must pay attention to when evaluating elderly gait and during interventions that aim to prevent the occurrence of falls.”